

# Causes of simulated, longterm changes in chlorophyll concentrations in the Baltic Sea

Jenny Hieronymus[1], Kari Eilola[1], Magnus Hieronymus[1], H. E. Markus Meier[2,1], and Sofia Saraiva[1]

[1]Research and Development Department, Swedish Meteorological and Hydrological Institute, Norrköping, Sweden
[2]Department of Physical Oceanography and Instrumentation, Leibniz Institute for Baltic Sea Research Warnemünde, Rostock, Germany.

*Correspondence to:* Jenny Hieronymus (jenny.hieronymus@gmail.com)

**Abstract.** The co-variation of key variables with modelled phytoplankton concentrations in the Baltic proper has been ex-
amined using wavelet analysis and results of a long-term simulation for 1850-2008 with a high-resolution, coupled physical-
biogeochemical circulation model for the Baltic Sea. By focusing on interannual variations it is possible to track effects acting
on decadal time scales such as temperature increase due to climate change as well as changes in nutrient input. The results
indicate the largest inter-annual coherence of phytoplankton with the limiting nutrient. However, after 1950 the coherence is
reduced due to high mixed layer nutrient concentrations diminishing the effect of smaller long-term variations. Furthermore,
the inter-annual coherence of mixed layer nitrate with riverine input of nitrate is much larger than the coherence between mixed
layer phosphate and phosphate loads. This indicates a greater relative importance of internal loads i.e. mixing of phosphate
from deeper layers. In addition, shifts in nutrient patterns give rise to changes in phytoplankton nutrient limitation. The mod-
elled pattern shifts from purely phosphate limited to a seasonally varying regime. The results further indicate some effect of
inter-annual temperature increase on cyanobacteria and flagellates. Changes in mixed layer depth affect mainly diatoms due to
a high sinking velocity while inter-annual coherence between irradiance and phytoplankton is not observed.
# 1 Introduction
The Baltic Sea is a semi-enclosed brackish water body separated from the North Sea and Kattegat through the Danish Straits.
It stretches from about $54^o$ to $66^o$ N and the limited water exchange with the ocean in the south gives rise to a large meridional
salinity gradient. The circulation is estuarine with a salty deepwater inflow from the ocean and a fresher surface outflow. The
Baltic Sea comprises a number of sub-basins connected by sills further restricting the circulation.
The limited water exchange and the long residence time of water have consequences for the functioning of the biology and
the biogeochemistry. The Baltic Sea is naturally prone to eutrophication and organic matter degradation keeps the deep water
oxygen concentrations generally low in between deep water renewal events. In turn, this leads to complex nutrient cycling with
different processes acting in oxygenized vs low oxygen environments.
The Baltic Sea has experienced anthropogenic pressure over the last century. After 1950 an intensive use of agricultural
fertilizer greatly enhanced the nutrient loads. Due to great improvements in sewage treatment the loads decreased again after
1980 (Gustafsson et al., 2012).



The intensification in nutrient loads led to an expansion of hypoxic bottoms (Carstensen et al., 2014). This has had effects on
the cycling of nutrients through the system. Anoxic sediments have lower phosphorus retention capacity resulting in increased
deep water phosphate concentrations. Thereby, the flux of phosphate to the surface intensifies even though the external loads
have decreased. Furthermore, as the anoxic area increases, the boundary between anoxic and oxic sediments where denitrifi-
cation occurs also increases. This results in a loss of nitrogen. Vahtera et al. (2007) described these processes as generating
a "vicious circle" where decreased DIN concentrations together with increased phosphate enhance the relative importance of
nitrogen fixation by cyanobacteria.
The importance of this coupling between oxygen and nutrients have been further examined in models. Gustafsson et al.
(2012) confirmed, using the model BALTSEM, that internal nutrient recycling has increased due to reduced phosphate retention
capacity, implicating a self sustained eutrophication where enhanced internal loads outweigh external load reductions.
In addition to the biogeochemical shifts in the Baltic Sea environment during the 20th century, sea surface temperatures have
increased (Siegel et al., 2006). This has an effect on the growth rate of phytoplankton as well as the speed of other biological
processes.
From satellite data, Kahru et al. (2016) detected a prolonged productive season as well as a chlorophyll maxima shifted
towards the maximum cyanobacteria concentration in July. The effect of temperature on the growth rate and stratification is
likely to have positively affected the strength of cyanobacteria blooms as well as the length of the growth season.
Schimanke and Meier (2016) analyzed multidecadal variations in Baltic Sea salinity and the coherence with different phys-
ical drivers. They used the wavelet transform to identify periodicities and wavelet conherency to analyse the driving mecha-
nisms.
In this paper we construct a thorough analysis of the co-variation of phytoplankton concentration with key variables that
have been affected by anthropogenic change over the 20th century. Using the biogeochemical model SCOBI (Eilola et al.,
2009; Almroth-Rosell et al., 2011) coupled to the 3d circulation model RCO we scrutinize the effect of nutrient loads, nutrient
concentration, temperature, irradiance and mixed layer depth on the modelled phytoplankton community.
The effect of anoxia on the nutrient limitation and on the primary production is complex. In addition to decreased phosphorus
retention capacity and denitrification, nitrification ceases in anoxic environments ultimately resulting in increased ammonium
concentrations (Conley et al., 2009). To elucidate the effect on the primary production, we calculate the degree of nutrient
limitation and its correlation with phytoplankton.
We have chosen to use a model run spanning 1850-2009. Thereby, we capture conditions relatively unaffected by anthro-
pogenic forcing as well as current conditions of eutrophication and climate change. Furthermore, we limit our investigation to
the Baltic Proper so as to capture relatively homogenous conditions with regards to the functioning of the biology. Our main
focus lies in inter-annual variations although some seasonal shifts will be investigated.



## 2 Methods

### 2.1 Study area

The Baltic Sea contains several different sub-basins with different characteristics in salinity and nutrient loads. We have here chosen to focus on the Baltic Proper. To obtain homogenous conditions we focus on the open ocean away from coasts. Areas where the depth is less than 20m are therefore removed. The study area is displayed in Fig. 1.

We have chosen to use a basin integrated approach. All variables have thus been horizontally integrated over the study area. This way we aim to gain an understanding of the overall functioning of the system.

### 2.2 Model

We have used a run with the model RCO-SCOBI spanning 1850-2009. RCO (Rossby Centre Ocean model) is a three-dimensional regional ocean circulation model(Meier et al., 2003). It is a z-coordinate model with a free surface and an open boundary in the northern Kattegat. The version used here has a horizontal resolution of 2nm with 83 depth levels at 3m intervals.

The Swedish Coastal and Ocean Biogeochemical model (SCOBI) (Eilola et al., 2009; Almroth-Rosell et al., 2011) is a one dimensional biogeochemical model that solves for three different water column and benthic nutrients (phosphate, nitrate and ammonia) as well as plankton functional types representing diatoms, flagellates and others (will be referred to as flagellates from here on) and cyanobacteria. Furthermore, the model contains nitrogen and phosphorus in one active homogenous benthic layer.

The model equations can be found in Eilola et al. (2009). Since we are exploring the effect of different variables on the growth of phytoplankton we will, for clarity, repeat some of them here.

The time rate of change of the concentration of phytoplankton chlorophyll in units of mg Chl m$^3$ day$^{-1}$ is described by

$$
\begin{aligned}
S_{PHY} = {} & GROWTH_{PHY} + NFIX + SINKI_{PHY} \\
& - SINKO_{PHY} - MORT_{PHY} - GRAZE_{PHY},
\end{aligned}
\tag{1}
$$

where subscript PHY stands for phytoplankton 1 (diatoms), 2 (flagellates) or 3 (cyanobacteria). GROWTH$_{PHY}$ describes the growth of phytoplankton, NFIX the production by nitrogen fixation, SINKI$_{PHY}$/SINKO$_{PHY}$ the flux of phytoplankton into/out of the current layer, MORT$_{PHY}$ the mortality and GRAZE$_{PHY}$ grazing by zooplankton.

The net growth of phytoplankton is described by the following expression,

$$
GROWTH_{PHY} = ANOX \cdot LTLIM \cdot NUTLIM_{PHY} \cdot GMAX_{PHY} \cdot PHY,
\tag{2}
$$

where ANOX is a logarithmic expression that approaches zero as the oxygen concentration becomes small. ANOX also contains a switch that sets it equal to zero when the oxygen concentration is zero so that no phytoplankton growth can occur in anoxic environments.





LTLIM expresses the phytoplankton light limitation and NUTLIM describes the nutrient limitation. Nutrient limitation
follows Michaelis-Menten kinetics where constant Redfield ratios are assumed in nutrient uptake. NUTLIM and LTLIM is
further described Sects. 2.2.1 and 2.2.2. GMAX is temperature dependent and describes the maximum phytoplankton growth
rate.
The difference between diatoms and flagellates are present in halfsaturation constants, maximum growth rate, temperature
dependence and sinking rate. Flagellates are more sensitive to a change in temperature than diatoms. Furthermore, the sinking
rate of diatoms is five times larger than that for flagellates.
The difference between cyanobacteria and the other phytoplankton species is more pronounced. Cyanobacteria can grow
either according to Eq. (2) or using nitrogen fixation according to
$$\text{NFIX} = \text{ANOX} \cdot \text{NF} \cdot \text{A3} \tag{3}$$
where NF is the rate of nitrogen fixation as a function of the phosphate concentration and temperature, and A3 is the concentra-
tion of cyanobacteria. Both NFIX and GROWTH of cyanobacteria is zero if the salinity is above 10. Furthermore, cyanobacteria
is the most temperature sensitive of the phytoplankton groups and no sinking velocity is assumed.
Other processes important for our results involves chemical reactions occurring in the water column or in the sediment.
Denitrification occurs in both the water column and the benthic layer and constitutes a sink for nitrate in case of anoxia.
Nitrification transforms ammonium into nitrate as long as oxygen is present. Phosphorus is adsorbed to the sediment and the
benthic release capacity of phosphate is a function of the oxygen concentration where more oxygen implies less release. The
phosphorus release capacity is also dependent on salinity where higher salinity means more phosphate is retained in the benthic
layer.

### 2.2.1   Nutrient limitation


Estimating nutrient limitation in nature is difficult. Usually this is done, either by comparing nutrient ratios to Redfield in eg.
the surface water or external supply or by some nutrient enrichment experiment (Granéli et al., 1990).
The idea of nutrient limitation as often used is based on that the primary production is directly limited by the nutrient
concentration in the ambient water and that the internal nutrient ratios in the phytoplankton are constant, i.e. in accordance with
a Redfield-Monod model (Redfield, 1958). However, cell-quota type models (Droop, 1973) are being increasingly implemented
and the use of constant internal nutrient ratios are becoming more and more questioned (Flynn, 2010).
Furthermore, N vs P limitation is a long standing debate. Tyrrell (1999) uses a box-modelling approach to show that in
steady state, nitrogen becomes slightly deficient while it is the external input and removal of phosphate that ultimately controls
the production.
Here, nutrient limitation is traditionally expressed assuming constand Redfield ratios and phytoplankton growth is limited
by either nitrogen or phosphate. The degree of nutrient limitation is described by:
$$\text{NUTLIM}_{\text{PHY}} = \min(\text{NLIM}_{\text{PHY}}, \text{PLIM}_{\text{PHY}}) \tag{4}$$



where $\text{NLIM}_{\text{PHY}}$ and $\text{PLIM}_{\text{PHY}}$ are the nitrogen and phosphate limitation respectively. In addition, $\text{NLIM}_{\text{PHY}}$ contains the
sum of the nitrate and ammonium limitation, i.e.
$$\text{NLIM}_{\text{PHY}} = \text{NO}_3\text{LIM}_{\text{PHY}} + \text{NH}_4\text{LIM}_{\text{PHY}}, \tag{5}$$
where
$$\text{NO}_3\text{LIM} = \frac{\text{NO3}}{\text{KNO3}_{\text{PHY}} + \text{NO3}} \cdot \exp(-\phi_{\text{PHY}} \cdot \text{NH}_4), \tag{6}$$
$$\text{NH}_4\text{LIM} = \frac{\text{NH4}}{\text{KNH4}_{\text{PHY}} + \text{NH4}}, \tag{7}$$
where NO3 and NH4 are the concentrations of nitrate and ammonium and $\text{KNO3}_{\text{PHY}}$ and $\text{KNH4}_{\text{PHY}}$ are the halfsaturation
constants for nitrate and ammonium respectively. The exponent in (6) represents preferential ammonium uptake (eg. Dortch
(1990); Parker (1993)).
$\text{PLIM}_{\text{PHY}}$ is modelled as,
$$\text{PO}_4\text{LIM} = \frac{\text{PO4}}{\text{KPO4}_{\text{PHY}} + \text{PO4}}. \tag{8}$$
Nutrient limitation is thus described by a number between 0 and 1 where 1 is no limitation. The constants $\text{KNO3}_{\text{PHY}}$,
$\text{KNH4}_{\text{PHY}}$ and $\text{KPO4}_{\text{PHY}}$ are the half saturation constants and differs between the different phytoplankton groups. The con-
stant $\phi_{PHY}$ in Eq. (6) determines the strength of ammonium inhibition of nitrate uptake. The values of the constants for each
phytoplankton type are given below.
$$KNO3_{\text{PHY}} = 0.5/0.25/0.25 \tag{9}$$
$$KNH4_{\text{PHY}} = 0.5/0.25/0.25 \tag{10}$$
$$KPO4_{\text{PHY}} = 0.1/0.05/0.05 \tag{11}$$
$$\phi_{PHY} = 1.5/1.5/1.5 \tag{12}$$
Note that the half-saturation constants for flagellates and cyanobacteria are equal which means that in absence of nitrogen
fixation, the nutrient limitation for the nitrogen fixing species is equal to that of flagellates.
In addition to the above given nutrient limitation of phytoplankton growth there exists a similar nutrient dependency on
nitrogen fixation. In the model this dependency reads
$$\text{NUTLIM}_{\text{NF}} = \frac{\text{aNFC}}{\text{aNFC} + \left(\frac{\text{NO}_3 + \text{NH}_4}{\text{PO}_4 \text{cNFC}} \text{dNFC}\right)} \cdot \frac{\text{PO}_4}{\alpha\text{NF} \cdot \beta\text{NF} + \text{PO}_4}, \tag{13}$$





where aNFC, bNCF, cNFC and dNFC are constants used for calculating the nitrogen fixation capacity which in turn is a
function of the ratio of inorganic nitrogen to phosphate. $\alpha$NF and $\beta$NF are constants determining the half-saturation for nitrogen
fixation. Again, NUTLIM$_{NF}$ approaches one if the nitrate and ammonium concentrations are zero and for large concentrations
of phosphate.

### 2.2.2 Effect of physical parameters

Changes in cloud-cover affect the incoming solar radiation and thereby the phytoplankton growth. The effect of light shows up
in the LTLIM term of Eq. (2).
$$\text{LTLIM} = \text{fracI}_{\text{PAR}}\text{I}_{\text{opt}} \cdot \text{EXP}\left(1 - \frac{\text{I}_{\text{PAR}}}{\text{I}_{\text{opt}}}\right), \tag{14}$$
$$\text{I}_{\text{PAR}}(z) = \alpha_{\text{PAR}}\text{I}_0 \cdot \text{EXP}(-\text{Kd} \cdot z) \tag{15}$$
$$\text{I}_{\text{opt}} = \max(\text{I}_{\text{opt,min}}, \alpha_{\text{opt}}\text{I}_0) \tag{16}$$
$$\text{Kd} = \text{Kd}_{\text{w}} + \text{Kd}_{\text{PHY}} + \text{Kd}_{\text{Y}} + \text{Kd}_{\text{D}} \tag{17}$$
$$\text{Kd}_{\text{PHY}} = \alpha_{\text{Kd}}(\text{A1} + \text{A2} + \text{A3}) \tag{18}$$
where I$_{\text{PAR}}$ is the photosynthetic available radiation and $I_{opt}$ is the optimum irradiance for phytoplankton growth. $I_{opt,min}$ is
a constant minimum optimum irradiance, I$_0$ is the surface irradiance and Kd is the vertical attenuation. Kd$_w$ is the background
attenuation, Kd$_{PHY}$ is the light attenuation due to the concentration of phytoplankton, Kd$_Y$ the attenuation due to humic
substances (calibrated) and Kd$_D$ the attenuation due to detritus. $\alpha_{Kd}$ is a constant vertical attenuation per unit chlorophyll.
A1/2/3 is the concentration of the respective phytoplankton type.
The mixed layer depth has been defined as the depth where a density difference of 0.125 kg m$^{-3}$ from the surface is reached
in accordance with what was previously done by e.g. Eilola et al. (2013). The density was calculated from modelled temperature
and salinity using the matlab routines by Jackett et al. (2006).

### 2.3 Forcing

The study use reconstructed (1850-2008) atmospheric, hydrological and nutrient load forcing and daily sea levels at the lateral
boundary as described by Gustafsson et al. (2012) and Meier et al. (2012). Monthly mean river flows were merged from
reconstructions done by Hansson et al. (2011) and by Meier and Kauker (2003) and hydrological model data by Graham
(1999), respectively. For further details about the physical model setup used in the present study the reader is referred to Meier
et al. (2016) and references therein.
The nutrient loads from rivers and point sources were (1970-2006) compiled from the Baltic Environmental and HELCOM
databases (Savchuk et al., 2012). Estimates of pre-industrial loads for 1900 were based upon Savchuk et al. (2008). The nutrient
loads were linearly interpolated between selected reference years in the period between 1900 and 1970. Similarly, atmospheric




loads were estimated (Ruoho-Airola et al., 2012). Nutrient loads contain both organic and inorganic phosphorus and nitrogen,
respectively.
Figure 2 shows the loads of Dissolved Inorganic Phosphorus (DIP) and Dissolved Inorganic Nitrogen (DIN) to the Baltic
Proper as used in the model. The loads are calculated from the runoff and annual mean nutrient concentrations (Eilola et al.,
2011). Thus the seasonal cycle in river loads is determined by the runoff. After a spin-up simulation for 1850-1902 utilizing
the reconstructed forcing as described above, the calculated physical and biogeochemical variables at the end of the spin-up
simulation were used as initial condition for 1850.
The open boundary conditions in the northern Kattegat were based on climatological (1980-2000) seasonal mean nutrient
concentrations (Eilola et al., 2009). The bioavailable fraction of organic phosphorus was assumed to be 100% in accordance
with the phosphorus supply from land runoff. Similar to Gustafsson et al. (2012) a linear decrease of nutrient concentrations
back in time was added assuming that climatological concentrations in 1900 amounted to 85% of present day concentrations
(Savchuk et al., 2008).
**2.4   The wavelet transform and wavelet coherence**
The continuous wavelet transform provides a method to decompose a signal into time-frequency space. In contrast to the
Fourier transform, the wavelet decomposition thus provides time localization and the means to see how periodicities change
with time. Wavelet coherence further expands the usefulness of the approach by allowing for calculating the time resolved
coherence between two time-series. For all wavelet calculations we use the Matlab wavelet package of described in Grinsted
et al. (2004), which is freely available at http://www.glaciology.net/wavelet-coherence.
In time-series with clear periodic patterns that is affected by environmental variables such as population dynamics and
ecology the benefits with this approach are significant (Cazelles et al., 2008). Several studys have implemented wavelet analysis
to plankton dynamics. Winder and Cloern (2010) applied the technique to time-series of chlorophyll-a from different localities
and discussed the annual and seasonal periodicities. Carey et al. (2016) discussed how the wavelet transform can be used to
track interannual changes in phytoplankton biomass and applied it to a 16-year time series of phytoplankton in Lake Mendota,
USA. In doing this they were able to identify periods when the annual periodicity was less pronounced. They discuss the benefit
of this technique in scrutinizing changes to the seasonal succession due to changes in external drivers.
The problem with the wavelet transform is that it requires a dataset without gaps. The time-series also needs to be sufficiently
long compared to the investigated periods. This makes it difficult to use the method to scrutinize the effect of processes acting
on longer time-scales, such as climate change, since long enough observational datasets are scarce. Hence, for our purpose
only a model based approach is feasible.
Here we use wavelet coherence to scrutinize the coherence between the three different phytoplankton groups (diatoms,
flagellates, and cyanobacteria) and nutrients, temperature, irradiance and mixed layer depth.



## 2.5  Observations

Oxygen and nutrient concentrations from the SCOBI model have been extensively evaluated against observations (Eilola et al., 2009, 2011, 2014) as well as other models (Eilola et al., 2011). Phytoplankton observations are more difficult to come by and our basin integrated approach makes it difficult to compare with observations from individual stations.

We have used a basin integrated dataset of monthly Chl-a for the Baltic Proper previously published in HELCOM (2012). The dataset includes all data from the Data Assimilation System (DAS) which is a database of Baltic Sea monitoring data hosted by the Baltic Nest Institute, Stockholm University, completed with data from the EUTRO-PRO project and HELCOM Indicator Fact Sheets (HELCOM, 2012). The surface layer was defined as the top 10m of the water column and coastal areas were removed.

## 3  Results and discussion

The model results shown are monthly means integrated over the basin. The different variables have also been vertically integrated over the mixed layer and/or from the mixed layer down to a depth of 150m. The first 20 yrs of the model run is exluded to minimize spinup effects.

We start out in Sect. 3.1 by scrutinizing the modelled concentration of phytoplankton and its seasonal cycle by comparison with observations. In Sect. 3.2, the coherence between nutrient loads and mixed layer nutrient concentrations as well as phytoplankton concentrations will be examined. Section 3.3 will consider the composition of nutrients and its effect on the phytoplankton concentrations. The effect of temperature and irradiance is scrutinized in Sect. 3.4 and in Sect. 3.5 the coherence of the mixed layer depth with phytoplankton is examined.

### 3.1  Phytoplankton - model and observations

Figure 3 shows the model results of basin integrated Chl-a concentration (the sum of the three different phytoplankton) over 0-10m together with the observations described above. The results are thus here integrated over a fixed depth rather than the mixed layer to better compare with the observations. The top panel of Fig. 3 displays observations and model results for the period 1990-2009. In order to illustrate the difference from pre-industrial, model results for the period 1880-1999 are also shown.

The top panel reveals that the largest values representing the spring bloom are underestimated in the model results compared to the observations. The model implements a constant C:Chl ratio of 50 in phytoplankton, while Jakobsen and Markager (2016) found that this ratio, in reality, varies throughout the year. The underestimation of the spring bloom in the model may therefore, at least in part, be explained by this simplified assumption. Furthermore, the wavelet transform reveals a strengthening in the model of the 6 month period relative to the annual compared to the early period (panel (c) and (d) in Fig. 3). This is caused by the large increase in cyanobacteria resulting in a stronger late summer bloom. The half year period is much weaker in the observations. In the upper panel of Fig. 3, this is visable as a greater observed difference between the spring and late summer





blooms. The smaller difference in magnitude between the two blooms in the model results reflects as stronger signal with a 6
month periodicity in the wavelet spectrum (panel (c) in Fig. 3).

## 3.2 Nutrient loads

To determine the effect of the riverine loads on the mixed layer nutrient concentrations we perform wavelet coherence. The
result is shown in Fig. 4. We have used riverine DIN and DIP loads in the results presented below. The use of instead total
bioavailable nutrient loads does not change the results.
The results show the clear annual cycle in riverine inputs and mixed layer nutrient concentrations. The phosphate loads show
little coherence on any other periodicity but DIN displays strong coherence on longer periods. Furthermore, there is a tendency
for a enhanced coherence during the later part of the run most likely caused by increased DIN loads.
The phase arrows on the annual scale points to the right during most of the run indicating that the seasonal peak in nutrient
loads and mixed layer concentrations are concurrent. However, during the period 1900-1920 the direction of the phase arrows
shifts upwards. This is a result of a persistent shift in the runoff maxima of about two months over the period. During this
period the peak in mixed layer nutrient concentrations thus precedes the runoff peak. The interpretation of this is not straight
forward but most probably it has to do with the scarcity of observations and the use of an integrated Baltic Sea runoff dataset.
To further investigate the lack of inter-annual coherence between riverine phosphate loads and mixed layer phosphate, the
wavelet coherence between mixed layer salinity and nutrients are examined and displayed in Fig. 5. Mixed layer salinity is
affected by freshwater input from land, precipitation, evaporation and mixing with deeper layers. The coherence spectrum
reveals higher coherence between mixed layer salinity and phosphate (top) on interannual periodicities than between salinity
and DIN (bottom). The coherence existing between salinity and DIN on periodicities longer than one year is antiphase i.e. low
salinity here coheres with high DIN concentrations. In contrast, the in-phase coherence between salinity and phosphate suggests
that the reason for the coherence is a greater importance of the internal source i.e. phosphorus release from the sediments that
eventually reaches the mixed layer through mixing with deeper layers.
Figures 6 and 7 show the coherence between the riverine input of phosphate/DIN and mixed layer chl concentrations of
diatoms (top), flagellates (middle) and cyanobacteria (bottom). There is again a strong annual coherence. There seems to be a
quite strong coherence between mainly diatoms and both nutrients on a 16 year period. However, given that the length of the
model run does not even give room for ten 16-year periods, this probably reflects the overall pattern of simultaneous increase
in riverine loads and chlorophyll concentrations over the second half of the 20th century.

## 3.3 Nutrients and nutrient limitation

We will here assess the coherence of nutrients with the phytoplankton concentrations. Furthermore, as described above, the
effect of nutrients on the primary production is controlled by the term NUTLIM, or degree of nutrient limitation, in Eq. (2).
We thus examine this term in and below the mixed layer. Even though there is no primary production in the deep water and
thus the nutrient limitation term has no effect here, a shift in the composition of nutrients in the deep water will affect also the



mixed layer. NUTLIM for the different plankton groups has been calculated offline from the monthly means according to Eqs.
(4) and (13).
Nitrogen has been shown to most often be limiting in the Baltic Proper, while phosphate is limiting in the northern basins
(Granéli et al., 1990; Tamminen and Andersen, 2007). However, our model results, displayed in Fig. 8, show phosphate limita-
tion for both diatoms and flagellates for the earlier part of the run. After 1980, seasonality appears in the mixed layer. Phosphate
is still limiting during winter while nitrogen becomes limiting after the spring bloom.
The extent of anoxic bottoms in the Baltic Sea has increased markedly over the past century. By compilation of a large
amount of temperature, salinity and oxygen observations Carstensen et al. (2014) found a 10-fold increase in the hypoxic area
since the beginning of the 20th century. They explained this to be primarily due to increased nutrient loads from land causing
increased deep water respiration but also due to increased temperatures resulting in reduced oxygen solubility.
In order to understand the limitation patterns found in our model run, we view the evolution of different nutrient concen-
trations. Figure 9 shows the anoxic volume together with the below mixed layer nutrient concentrations. In conjunction with
the increased anoxic volume we find a clear increase in ammonium concentration. This is due to a decrease in nitrification
and is seen also as a decrease in the nitrate concentration. Furthermore, expanding anoxic bottoms increase the boundary area
between anoxic and oxic water where denitrification occurs resulting in a further loss of nitrate.
Figure 9 also shows that the phosphate concentration increases from the mid 20th century through the rest of the model run.
This is a combined effect of increased riverine loads and enhanced sedimentary release due to anoxia.
The mixed layer displays corresponding patterns of increased phosphate and decreased nitrogen (Fig. 10). The seasonal
variations are however much greater since the majority of the primary production occurs here and since the mixed layer is
directly affected by riverine input. The mixed layer also comprises a smaller volume of water. Despite quite high wintertime
concentrations, the spring bloom almost completely depletes the nitrogen. The seasonality that appears after 1980 in mixed
layer nutrient limitation with nitrogen limitation after the spring bloom is thus a results of the larger relative increase in
phosphate compared to nitrogen.
The sum of the effects on the nutrient concentrations shows up in the nutrient limitation expressions (Eqs. (5)-(8)).
The evolution of NUTLIM in the surface layer and the deep water for the three phytoplankton is shown in Fig. 11. There is
a clear increase over the 20th century and a shift towards less limited conditions.
After 1980 there is a shift in the variability of nutrient limitation for diatoms and flagellates most clearly visible in the deep
water. This shift is also visible in the lower two panels of Fig. 8 which show that deepwater NUTLIM shifts towards a purely
nitrogen limited regime while NUTLIM for flagellates mostly display a seasonal pattern. The lower variability is due to the
characteristics of the nitrogen limitation Eq. (5). The concentrations of nitrate and ammonium at the end of the model run
corresponds to a minimum in Eq. (5). Therefore, even though the concentrations change, NUTLIM changes very little.
To see how the phytoplankton concentrations are connected to nutrient concentrations and nutrient limitation, we continue
by scrutinizing the wavelet coherencies.
Figure 12 and 13 show the wavelet coherence between mixed layer phosphate and DIN and phytoplankton. Diatoms which
are the most nutrient limited group show strong inter-annual coherence with phosphate during the first, consistently phosphate



limited part of the run (see Fig. 8). During the later part of the run the nutrient and phytoplankton concentrations are so high
that smaller inter-annual variations have little effect.
Since nitrogen limitation only occurs after 1980 and after the spring bloom and thus only affects the much smaller diatom
and flagellate autumn blooms no coherence between phytoplankton and nitrogen is visable in Fig. 13.
To further illustrate the shift from the more nutrient limited regime of the first part of the run we calculate the wavelet
coherence between NUTLIM for the different phytoplankton and the result is displayed in Fig. 14. Again, diatoms show strong
coherence during the first, more nutrient limited part of the run.
In Fig. 15 we calculate the wavelet coherence between below mixed layer NUTLIM and the three types of phytoplankton.
Again, the coherence spectrum shows the most inter-annual coherence for the more nutrient limited diatoms. However, the
phase arrows display some interesting features. After 1980 the phase arrows within the annual coherence period change direc-
tion. This occurs both for diatoms where they shift from downward, indicating that the annual NUTLIM periodicity precedes
the annual diatom periodicity by 90 degrees, i.e. 3 months, to upwards, instead indicating that the diatoms precedes NUTLIM.
A similar pattern is visable also in flagellates.
To investigate the reasons for this, we have plotted the month of maximum NUTLIM in Fig. 16. The figures show a clear
shift occuring after 1980 correlating with a strengthening of cyanobacteria blooms. The deep water changes its maxima to the
late summer months while a slight shift from February to March is apparent for diatoms. Mixed layer NUTLIM for flagellates
displays no clear shift.
Figure 17 shows the timing of the maximum chlorophyll concentration for the different phytoplanktons as well as their sum.
Flagellates displays a weak shift towards May after 1960 but no other shifts are visible in the individual phytoplankton types.
However, the total chlorphyll concentration (Diatoms + Flagellates + Cyanobacteria) displays a few years at the very end of the
run where the chlorophyll maximum corresponds to the maximum for cyanobacteria. From satellite data, Kahru et al. (2016)
found a similar shift in chorophyll maximum from the spring bloom in May to the cyanobacteria bloom in July.

### 3.4   Temperature and irradiance

The mixed layer temperature has increased over the 20th century. Figure 18 shows the 2-yr moving average of mixed layer
temperature. To scrutinize the effect of temperature on the concentration of phytoplankton, the wavelet coherence between
temperature and phytoplankton have been plotted in Fig. 19. The results suggest that the temperature increase after 1990 might
have had an effect on cyanobacteria and flagellates. It is also noticable that the temperature increase observed between 1900
and 1940 probably had an effect on cyanobacteria. This is also in agreement with the model formulation where cyanobacteria
are the most sensitive to temperature followed by flagellates.
Light impacts primary production through the term LTLIM in Eq. (2). However, irradiance display very little variation on
any other periodicity than the annual as can be observed in a wavelet power spectrum (not shown). Therefore there exists
almost no coherence between phytoplankton and irradiance apart from the annual and semiannual.



## 3.5 Mixed layer depth

The lower panel of Fig. 18 shows the two year moving average of mixed layer depth averaged over the basin. We calculate the coherence between mixed layer depth and diatoms, flagellates and cyanobacteria in Fig. 20.

Apart from the annual cycle there is a strong coherence between mixed layer depth and diatoms, and to some extent flagellates, on shorter periodicities as well. That is, the concentration of diatoms residing in the mixed layer seems to covary quite well on periodicities equal to or shorter than one year. The model value for diatom sinking rate is five times higher than that for flagellates while cyanobacteria is assumed to have no sinking rate. In a shallow mixed layer the diatom concentration decreases faster than in a deep mixed layer because of the large sinking rate. In the wavelet coherence spectrum we thus see in-phase short term coherence.

## 4 Summary and conclusions

With a main focus on inter-annual variations, the coherence of the mixed layer concentrations of phytoplankton with key variables affecting the primary production has been examined for the Baltic Proper.

Riverine input of nutrients is an extremely important variable in the Baltic Sea and the large increase during the 20th century has initiated spreading of anoxic bottoms (Carstensen et al., 2014). We found quite strong coherence between riverine input of DIN and mixed layer DIN but not a similar relationship between riverine phosphate input and the corresponding mixed layer concentration. As mixed layer salinity displayed in-phase inter-annual coherence with phosphate and only weak anti-phase coherence with DIN we conclude that this is most probably due to a greater importance of the internal source of phosphate from lower layers.

We further found that the pattern of nutrient limitation in and below the mixed layer have changed in the model since 1980. Below the mixed layer, the limitation pattern changes from phosphate to nitrogen for diatoms and to seasonally shifting between phosphate and nitrogen. Within the mixed layer, the pattern changes from pure phosphate limitation to seasonally shifting for both diatoms and flagellates. This is due to decreased deep water oxygen concentrations and a rapid expansion of anoxia after 1970. The phosphate concentrations increase due to enhanced sedimentary release, denitrification results in loss of nitrate and reduced nitrification decreases the transformation of ammonium to nitrate. The combined effect results in nitrogen limitation after the spring bloom which benefits cyanobacteria.

The mixed layer concentrations of nutrients affect the primary production in the model through the nutrient limitation term, NUTLIM. The phytoplankton group most strongly limited by nutrients in the model is diatoms. The connection between primary production and the nutrient limitation term is visable as a strong inter-annual coherence between diatoms and phosphate as well as NUTLIM before 1940. After 1940 NUTLIM as well as the concentrations of the individual phytoplankton species has gained such high values that smaller inter-annual variations have little effect on the production. Similarly, the less nutrient sensitive group flagellates shows much smaller inter-annual coherence with phosphate even before 1940. NUTLIM for this group is already high enough so that small long-term variations do not reflect strongly in the results.





Very little inter-annual coherence is visable also between phytoplankton and nitrogen. The spring bloom is phosphate limited
throughout the run except for a few years after 1990 where diatoms display nitrogen limitation. The much weaker diatom and
flagellate autumn bloom displays no inter-annual coherence with DIN most likely due to the high NUTLIM levels.
The shift in nutrient limitation patterns is also visable in a slight forward shift in the month of maximum mixed layer
NUTLIM for diatoms after 1980, although a similar shift cannot be seen for flagellates. Below the mixed layer, maximum
NUTLIM shifts significantly towards late summer for both diatoms and flagellates. Furthermore, the annual maximum of total
chlorophyll concentration (Diatoms + Flagellates + Cyanobacteria) displayed a few years at the end of the run where the
maximum corresponded to the autumn bloom due to the large increase in cyanobacteria. This is in agreement with Kahru et al.
(2016) who found from satellite data that the annual chlorophyll maximum has shifted from the spring bloom maximum in
May to the cyanobacteria bloom in July.
The mixed layer temperature in the Baltic Proper has increased during the 20th century. We found some response of this
mainly from the most temperature sensitive phytoplankton group cyanobacteria during periods of large interannual temperature
increases. Flagellates, being more temperature sensitive than diatoms, seems to display a coherence with the temperature
increase occuring after 1980.
Variations in mixed layer depth affects mainly diatoms as these have a high sinking speed. In-phase coherence on periodic-
ities shorter than one year indicates that large seasonal changes in the mixed layer depth significantly affects the mixed layer
concentrations while smaller interannual variations are of little consequence.
Finally, the effect of irradiance on primary production was scrutinized. However, interannual irradiance variations have very
little effect on the primary production.
In conclusion, interannual variations have affected the primary production mostly through the limiting nutrient phosphate
before 1950 in our model run. After that nutrients and phytoplankton exists in the water column at such high concentrations
that smaller interannual variations have much less effect. Furthermore, the mixed layer concentrations of DIN show strong
interannual coherence with riverine DIN input while riverine phosphate displays almost no coherence with the corresponding
mixed layer concentration. Instead, in-phase coherence with mixed layer salinity indicates a stronger importance of mixing
with lower layers. Expanding low oxygen conditions in the deep water has resulted in a shift in the composition of nutrients. In
the model, this results in seasonality in the nutrient limitation pattern of the mixed layer with phosphate limitation in the spring
and nitrogen limitation after the spring bloom.
## 5  Data availability
The model data on which the results in the present study are based on are stored and available from the Swedish Meteorological
and Hydrological Institute. Please send your request to ocean.data@smhi.se.
*Acknowledgements.*  This work was funded by the Swedish Research Council (VR) within the project " Reconstruction and projecting Baltic
Sea climate variability 1850-2100" (Grant 2012-2117).



Funding was also provided by the Swedish Research Council for Environment, Agricultural Sciences and Spatial Planning (FORMAS)
within the project "Cyanobacteria life cycles and nitrogen fixation in historical reconstructions and future climate scenarios (1850-2100) of
the Baltic Sea" (grant no. 214-2013-1449). The study contributes also to the BONUS BalticAPP (Wellbeing from the Baltic Sea - applications
combining natural science and economics) project which has received funding from BONUS, the joint Baltic Sea research and development
programme.
This research is also part of the BIO-C3 project and has received funding from BONUS, the joint Baltic Sea research and development
programme (Art 185), funded jointly from the European Union's Seventh Programme for research, technological development and demon-
stration and from national funding institutions.
We thank Bärbel Muller-Karulis for providing the observational data.





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





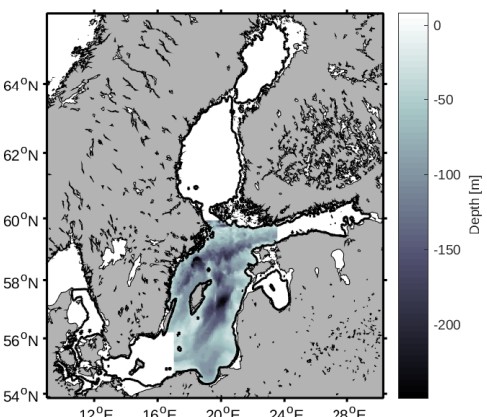

**Figure 1.** Study area. The grey scale represents depth in m.

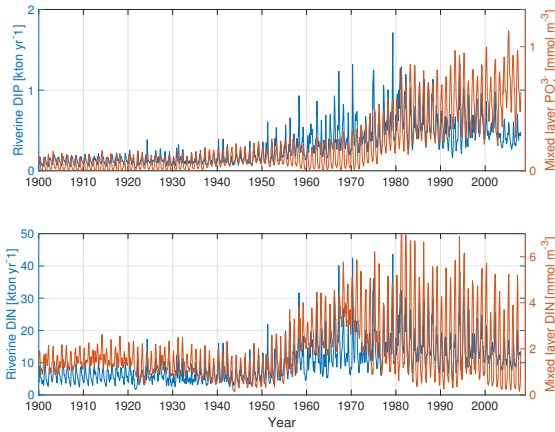

**Figure 2.** The top panel shows riverine phosphate loads (blue) and mixed layer concentration of phosphate (red) and the bottom panel shows riverine DIN (blue) and mixed layer DIN (red).




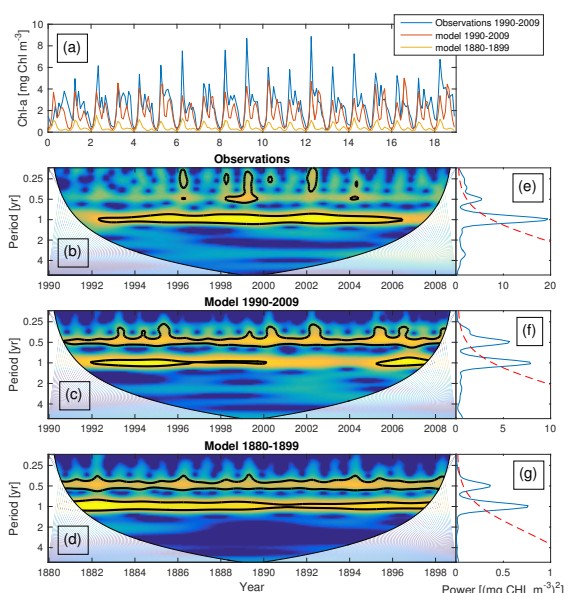

**Figure 3.** Modelled basin integrated chlorophyll compared to observations. (a) shows observations (blue) and model results (red) for the period 1990-2009 together with model results for the period 1880-1999 (yellow). The lower three panels shows the wavelet spectra for (b) observations, (c) model results for 1990-2009 and (d) model results for the period 1880-1899. The y-axis shows the periodicity and the colors represent the wavelet power. The black curves in the wavelet figures represent the 95% confidence level relative to an AR1 spectrum. (e), (f) and (g) show the corresponding global power spectra together with the AR1 spectrum (red). The white areas in the wavelet figures represents the cone of influence in which the results are impacted by edge-effects and are therefore not shown.




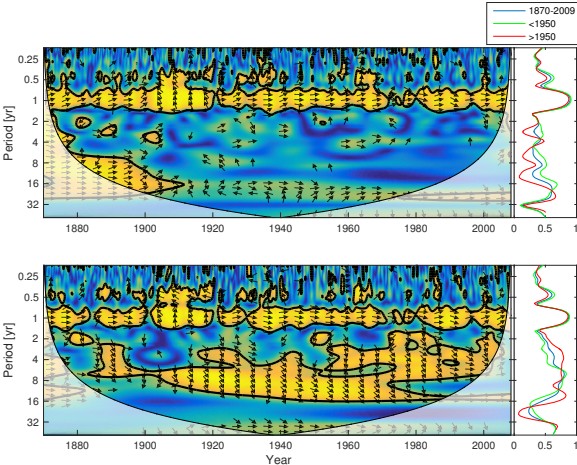

**Figure 4.** Wavelet coherence between riverine phosphate and mixed layer phosphate concentration (top) and riverine DIN and surface DIN concentration (bottom). The arrows indicates the phase lag. When pointing to the right the two time-series are in phase and when pointing in the opposite direction anti-phase. The right panels show the integrated coherence for the whole period (blue) and before (green) and after (red) 1950.

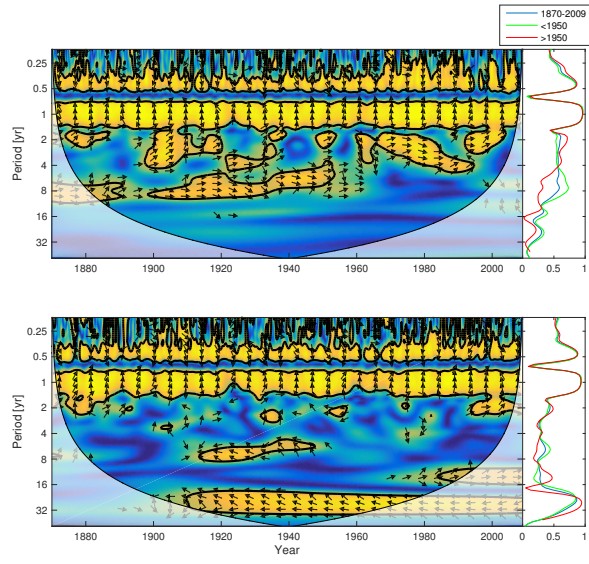

**Figure 5.** Wavelet coherence between mixed layer salinity and phosphate concentration (top) and mixed layer salinity and nitrate concentration (bottom). The right panels show the integrated coherence spectrum.





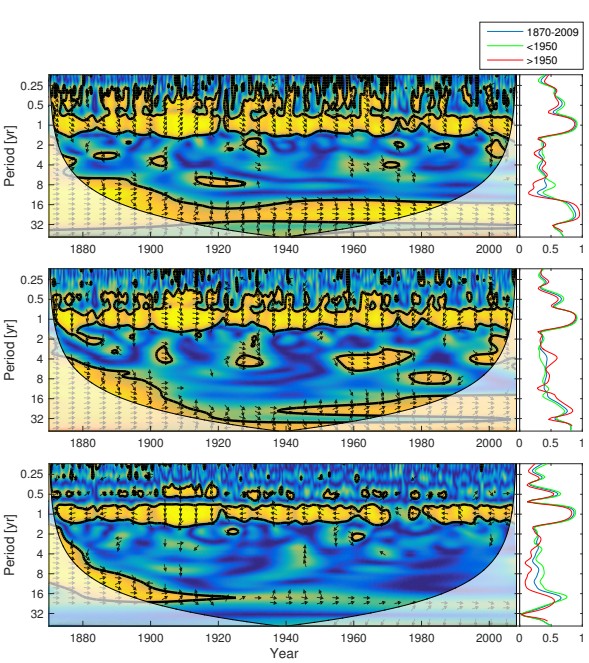

**Figure 6.** Wavelet coherence between riverine phosphate and diatoms (top), flagellates (middle) and cyanobacteria (bottom). The right panels show the integrated spectrum.

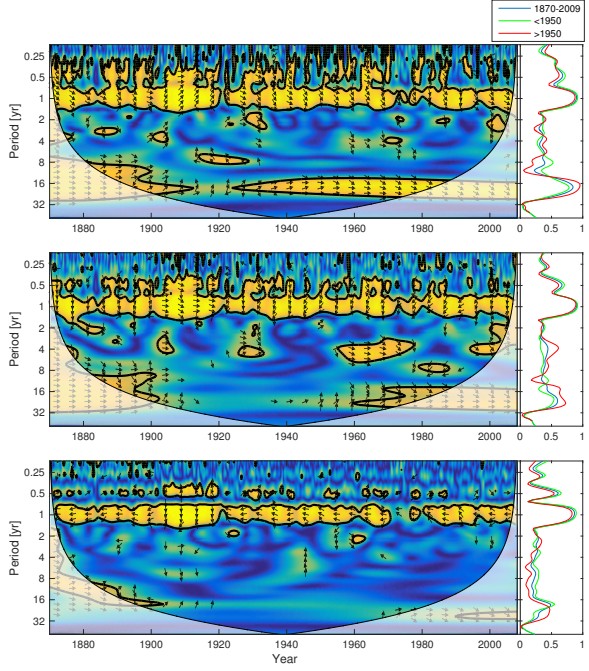

**Figure 7.** Wavelet coherence between riverine DIN and diatoms (top), flagellates (middle) and cyanobacteria (bottom). The right panels show the integrated spectrum.

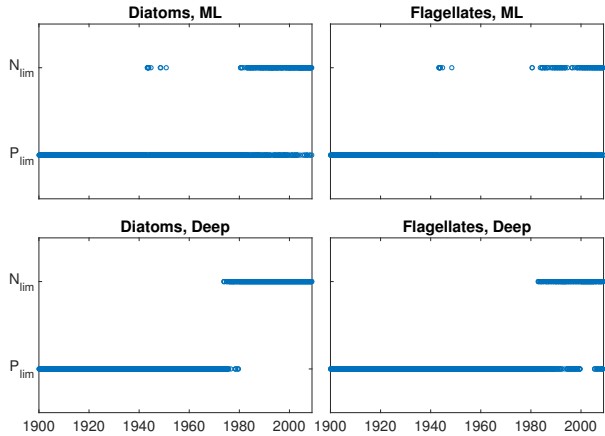

**Figure 8.** Nitrogen or phosphate limitation as function of time in the mixed layer (upper panels) and in the deep water (lower panels) of diatoms (left panels) and flagellates (right panels).





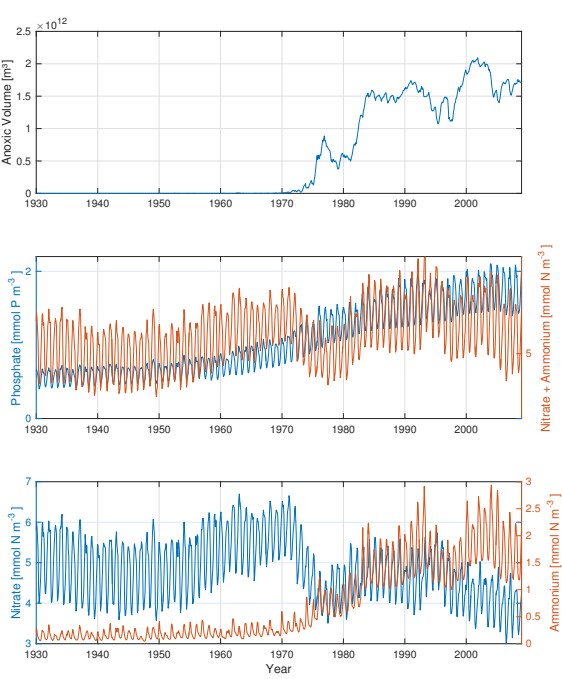

**Figure 9.** Time-series of anoxic volume (top), below mixed layer concentrations of phosphate (blue) and DIN (nitrate + ammonium, red) (middle) and nitrate (blue) and ammonium (red)(bottom).



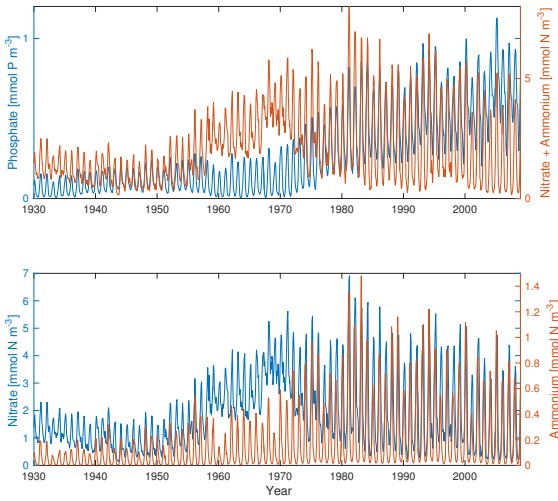

**Figure 10.** Time-series of mixed layer phosphate (blue) and DIN (nitrate + ammonium, red) concentration (middle) and nitrate (blue) and ammonium (red) concentration (bottom).

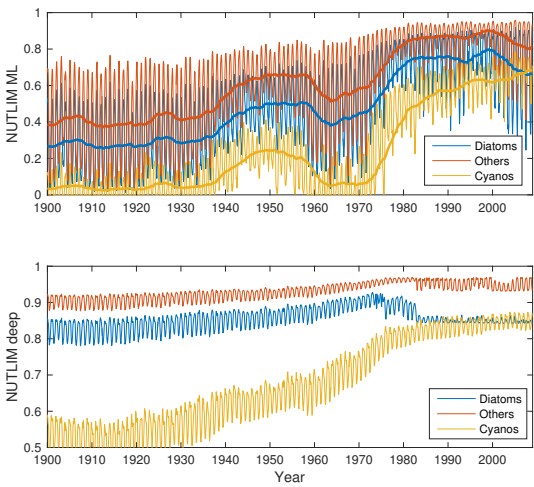

**Figure 11.** Time-series of nutrient limitation in the mixed layer (top) and below (bottom) for diatoms (blue), flagellates (red) and nitrogen fixation (yellow). The thicker lines in the top panel show the 5yr moving average.





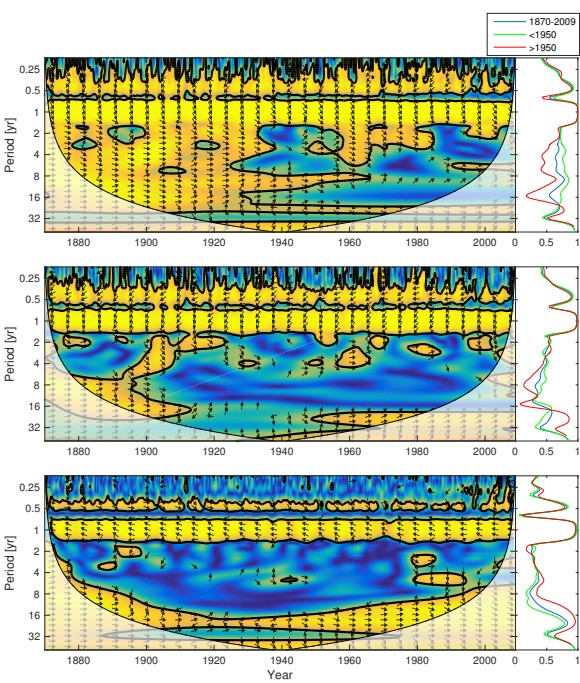

**Figure 12.** Wavelet coherence between mixed layer phosphate concentration and diatoms (top), flagellates (middle) and cyanobacteria (bottom).





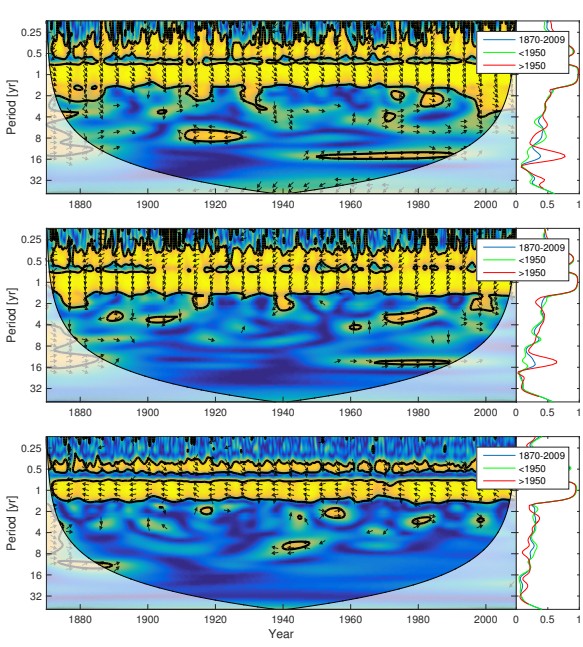

**Figure 13.** Wavelet coherence between mixed layer DIN concentration and diatoms (top), flagellates (middle) and cyanobacteria (bottom).



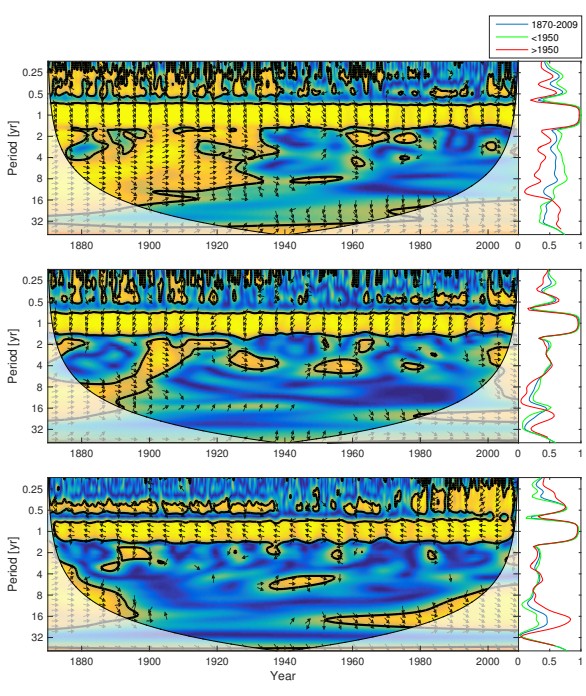

**Figure 14.** Wavelet coherence between mixed layer NUTLIM and diatoms (top), flagellates (middle) and cyanobacteria (bottom).





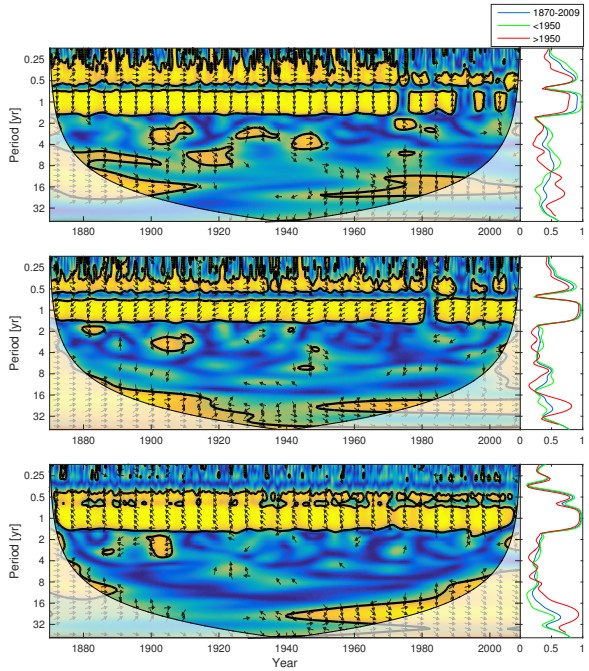

**Figure 15.** Wavelet coherence between deep water NUTLIM and diatoms (top), flagellates (middle) and cyanobacteria (bottom)

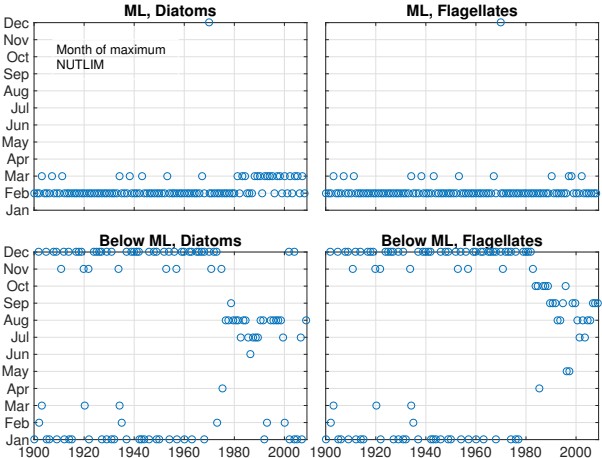

**Figure 16.** The month of maximum NUTLIM for diatoms (left) and flagellates (right) in the mixed layer (top) and below (bottom).





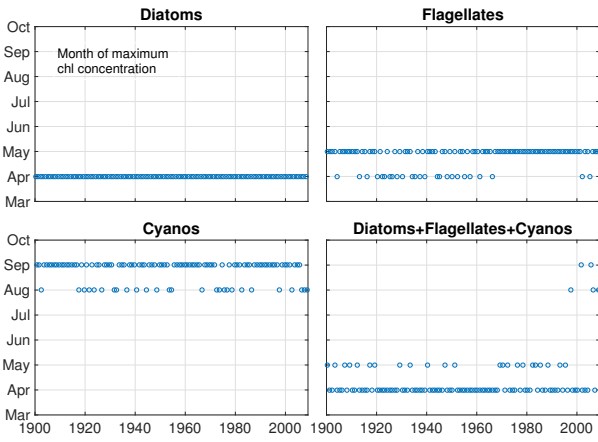

**Figure 17.** The month of maximum concentration of diatoms, flagellates and cyanobacteria as well as their sum.

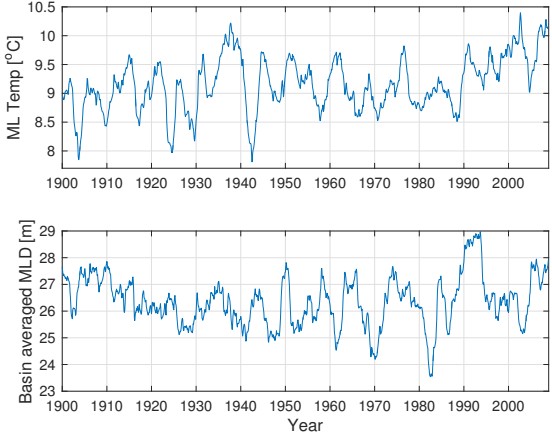

**Figure 18.** 2-yr moving average of mixed layer temperature (top) and mixed layer depth (bottom).





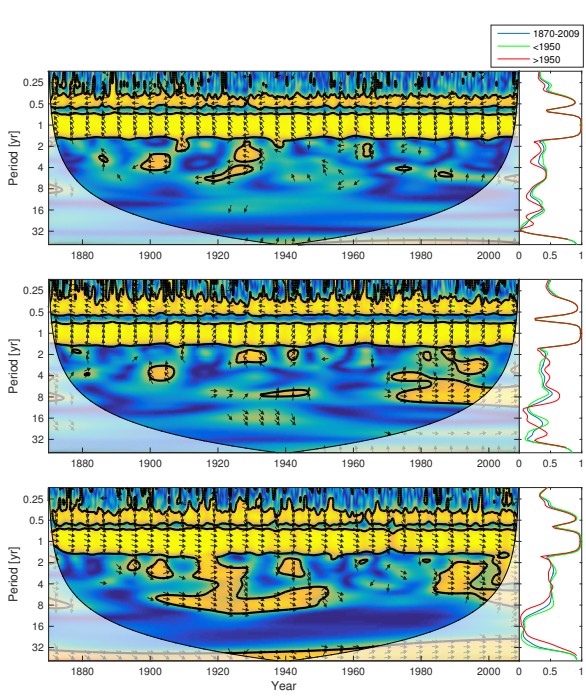

**Figure 19.** Wavelet coherence between mixed layer temperature and diatoms (top), flagellates (middle) and cyanobacteria (bottom).





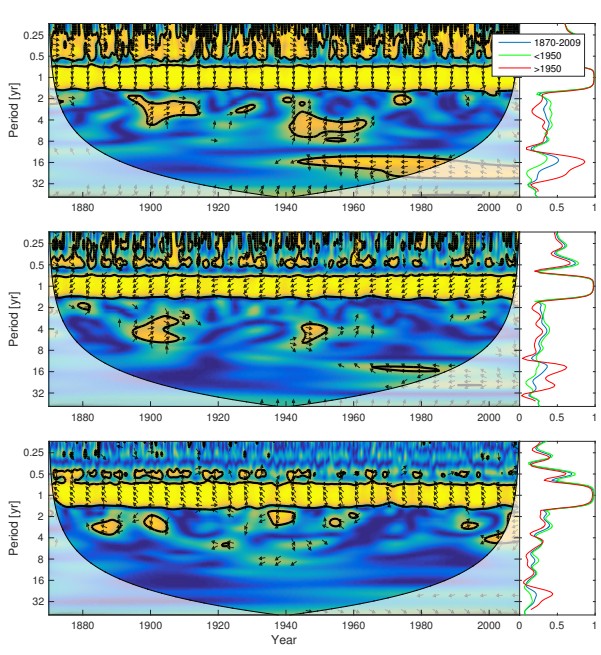

**Figure 20.** Wavelet coherence between mixed layer depth and diatoms (top), flagellates (middle) and cyanobacteria (bottom).