# Peer review of "Causes of simulated long-term changes in phytoplankton biomass in the Baltic Proper: A wavelet analysis"

_Biogeosciences, 2017_

## Referee Comment (RC1) · Anonymous Referee #3 · 24 May 2017

The manuscript present interpretations of long-term ecosystem model results by wavelet analysis. The method to understand the model behaviour applied here is relatively new and promising. Generally, the manuscript is well written.

Some open questions and suggestions:

Original SCOBI model equations were published in Eilola et al. 2009. To my opinion introduction to SCOBI model is too long and could be easily shortened in a new publication. Authors reproduce already published part of the model equations and description. However, it takes more space than in the original Eilola et al. 2009 paper. Moreover, text and equations are slightly different from original publication, that is confusing. Previews publication contains number of reference to sources of SCOBI model equations (see Eilola 2009, Table A.4.). These references are not listed in the current

version of model description. Without these references it is hard to follow why particular formulation is relevant.

Already in the abstract combination of words "mixed layer 'parameter' concentrations" appears as solid term. However I did not find in the text how it was defined. Is it mean value of horizontal mean 'parameter' in horizontal mean mixed layer ? or it is integrated characteristic ?

Salinity in the Baltic Sea and in the Baltic Proper have strong lateral gradient. However, mixed layer depth (MLD) was defined as constant density difference. Could it be that with decrease of salinity MLD will increase ? Could it be that seasonal variability in surface effects MLD and at the end all results ? The part with mixed layer definition should be extended and some how emphasized. May be it makes sense to include it as additional subsection.

The "basin integrated approach" was used here (line 61). Would be good to see in the text why this is acceptable (preferably in more than one sentence, line 62).

While SCOBI model is 1D model (line 67), I would suggest to show results of wavelet analysis for idealized 1D cases. So it could be seen how certain changes are reflected in final results of wavelet analysis. For my opinion such sensitivity test could enhance conclusions. Otherwise, section 2.4 should be extended with some aspects of wavelet coherence.

Analysis focuses mainly on river loads and its changes. Other nutrient sources like atmospheric deposition, exchange with other Baltic Sea regions and there possible effect should be mention somehow.

It could be considered to include wavelet analysis in to the title – to my opinion application of this method is among the most interesting aspects of this manuscript.

Line 75: eq. 1. NFIX is nitrogen fixation term, in all phytoplankton groups it looks strange. Is it a misprint? Line 78: SINKIphy / SINKOphy is it sinking of phytoplankton

[Figure]

? Line 148: eq. 14. : "frac" is a misprint. Lines 177 - 181: Paragraph is confusing. It starts with sentence about open boundary, but last two sentences are probably about river loads. Please specify in more details: what these assumption were applied to?

―――――――――――――――

---

## Referee Comment (RC2) · O.P. Savchuk (Referee) · 27 May 2017

Reviewer comments to manuscript by J. Hieronymus, K. Eilola, M. Hieronymus, H.E.M. Meier, and S. Saraiva "Causes of simulated, longterm changes in chlorophyll concentrations in the Baltic Sea" submitted to "Biogeosciences"

The study deals with the application of statistical wavelet analysis to the results of numerical simulation of multi-decadal ecosystem dynamics performed on a 3D biogeochemical model. This combination of empirical and theoretical approaches is rather novel in both methodological and, especially geographical aspects and could be interesting not only to the readers of, say, "Ecological modelling" but also to a much wider audience of "Biogeosciences". However, in order to reach this audience the scientific presentation and analysis should be significantly streamlined, deepened, and made

much more relevant to the Baltic Sea realities. To my mind, the necessary efforts would amount to a moderate or even major revision.

1. General comments and suggestions 1.1 Objectives of the study As can be understood from the title, the major goal of this study is to find (reveal, explain) the "causes" that determine the simulated long-term dynamics of phytoplankton in the Baltic Sea. Meanwhile, all the causal relationships driving the variations of variables had already been assumed and explicitly parameterized in the model formulation and algorithms, including prescription of the initial and boundary conditions. Correspondingly, the simulated variations is merely a result of the dynamical balance between positive ("sources") and negative ("sinks") terms ("fluxes") in a system of differential equations. In that sense, the causes are already implicitly known and all that is needed is just a clever quantitative analysis of the fluxes and balances that determine the dynamics. Such kind of analysis has already been successfully performed and published in numerous papers, including those co-authored by the authors of both this manuscript and this review. Therefore, the necessity in empirical approach to deterministic causal relationships needs a specific motivation and justification that must be given already in the Introduction.

1.2 Interpretation of results To my mind, there are two major challenges to this study and presentation of its results. Firstly, it should always be remembered by authors themselves and made clear for the reader where and when you are discussing the results of simulation vs. where and when – the real Baltic data and conditions. As it seems to me, the text is often written in such a way as if this distinction is almost forgotten (neglected). Just a few, by far non-comprehensive, examples: "The co-variation of key variables with modeled phytoplankton… (line 1)", "…the effect of nutrient loads, nutrient concentration, temperature, irradiance and mixed layer depth on the modeled phytoplankton community (lines 46-47)", whereas part of these "variables" is prescribed, while another part is MODELED similarly to phytoplankton; at lines 341-342 "the coherence of the mixed layer concentrations of phytoplankton with key variables

affecting the primary production has been examined for the Baltic Proper" at this start of Summary and conclusions you must explicitly indicate that the entire analysis was made on results of modelling. From that perspective, the entire text should be carefully read and appropriately edited.

Secondly, interpretations of the wavelet spectra in time-frequency domain, especially interpretation of the wavelet coherence must be explained already in Methods in more detail in respect to what does it show – periodicities and their coincidences, time lags, phase shifts, correlation and its strength, what is a wavelet power, what are the AR1 and global power spectra, etc.? Particularly important are considerations and interpretations involving the "coincidence" vs. "causality", i.e. "simultaneously (coherently) occurring" (by the same or even different reasons) vs. "because of". The text is often written as if you imply the latter. It could be very helpful if you would illustrate your explanations with the wavelet analysis of water temperature as more reliably simulated.

1.3 Chlorophyll as a measure of the phytoplankton biomass As is well known, the C:Chl ratio in the phytoplankton of temperate latitudes varies from 10 – 20 in the winter to over 100 in the summer but the exact seasonal patterns and ranges of these variations are different between both the algae species (functional types) and sub-basins. Therefore, the use of Chl as a model variable with the constant C:CHL=50 may be considered just as a some nominal measure of the phytoplankton biomass, which comparability to real measurements has a large inherent uncertainty. Apparently, the authors understand this conventionality very well as, for instance there is no mentioning of chlorophyll in the Abstract at all and quite a few in the Summary and conclusions. Such understanding should be made clear for the reader, while the word "chlorophyll" has to be replaced with "biomass" or "phytoplankton biomass" wherever it is possible.

1.4 Targeted scales As the seasonal cycle is only expected for both the functioning of the Baltic Sea ecosystem and variations of the prescribed boundary conditions, the authors have to focus much more on a longer, interannual to decadal time scales, reverting to the annual and shorter scales if only absolutely necessary for important

discoveries.

1.5 Relation to the real world There are almost no comparisons to the data and estimates based on measurements and experiments. Then, it should be explained why an exception was made for chlorophyll (Section 3.1), in contrast, for instance, to other, also simulated variables like temperature, salinity, and nutrients. The revision choice could be between either repeating similar comparisons and wavelet analysis for other measured variables, thus expanding the entire study and shifting it towards model validation, or excluding observed chlorophyll from Section 3.1 entirely, thus confining the analysis to merely simulated time-series. Taking into account my comment 1.3, I would recommend the latter, while a relation to the real world could rest solely on the literature references.

1.6 Mode of presentation To my mind, there are two major flaws in how the manuscript is written and results are presented. In addition to a lack of comparison to the real world data, there are almost no references to published studies and conclusions that are pertinent to findings and features that are presented and discussed in the manuscript. Moreover, in its present form the manuscript looks rather as a technical report, kind of monotonically listing some results calculated just because there is a novel tool and there are computed variables. As I have already started indicating above, the manuscript should be made more conforming to the usual scientific standards, that is highlighting: WHY (which yet unsolved problems, justification of the approach to solve them), HOW (pros and contras of the tool and processed material, including plausibility of simulation), WHAT (the novel conclusions, how realistically and reliably they are comparing to existing knowledge and views, uncertainty of results, unsolved remnants).

2. Specific comments and suggestions 2.1 Title According to my comments above, an every word in the title should be carefully reconsidered starting from "causes" (are you revealing deterministic causes or just interesting co-variations?) to "chlorophyll" (phytoplankton biomass) to the Baltic Sea (without Arkona and Bornholm basins your

area is not even the Baltic Proper). An explicit indication already in the title at the implemented wavelet analysis or even wavelet coherence would be appropriate as well.

2.2 Abstract The "internal loads" here and elsewhere is a bad term for a reversible phosphate exchange between the water body and sediments where the total pool have been accumulating for decades if not for centuries, because of external loads. Please, reconsider everywhere.

2.3 Introduction Besides the general description of the scene, it needs a better emphasis on the yet unanswered scientific questions as a prelude to better motivation and justification of the necessity in empirical wavelet analysis.

2.4 Methods Within a basin-integrated approach you are actually dealing not with the "horizontally integrated" values (that must then be in thousand tonnes) but the "horizontally averaged" concentrations, biomasses, depths, etc. or basin-averaged as at line 332. Correct, please, at lines 61, 205, 211, 220, 221, Fig. 3, and elsewhere, wherever I could have missed it.

2.4.1 Model – I understand your reasoning at lines 72-73 but suggest to carefully reconsider which explanations you want to give already here, in Methods, i.e. pretty far away from their subsequent usage in Results and Discussion and which reminding would be enough to make directly there. Just as example,"The model value for diatom sinking rate is five times higher than that for flagellates while cyanobacteria is assumed to have no sinking rate" at lines 336-337 is quite enough and even more informative than "Furthermore, the sinking rate of diatoms is five times larger than that for flagellates" at lines 90-91 and about blue-greens at line 97. – Also, I am not sure how necessary is Eq.1 without indication of conversions between Chl and C, explanations on sinking terms and on absence of NFIX in equations for diatoms and flagellates. Perhaps, more important is the explanation about Chl as a measure of phytoplankton biomass and a constant C:Chl and C:N:P ratios. The decision could depend on whether you need to

refer to Eq.1 in subsequent analysis. – In Eq. 3, NF is not defined anywhere. I guess, it could be a product of Eq. 13 and some function of temperature but it is not presented. – "...higher salinity means more phosphate is retained in the benthic layer..." (lines 102-103). In my parameterization, borrowed also by Eilola et al. (2009, p. 168) it is the other way round – less retention (higher release) at higher salinity. Correspondingly, check, please, at lines 251-253 and elsewhere. – Your parameterization of the nitrogen limitation (Eq. 5) as a sum of separate/independent (!?) ammonium and nitrate limitations (Eqs. 6 and 7) under certain conditions is higher than 1, hence, contradicts to the basic assumption $0 < NLIM < 1$ and amplifies the growth rate rather than limits it. As can be calculated by these equations with constants (9, 10, 12) and the real Baltic Proper monthly averages of ammonium and oxidized nitrogen concentrations, the value of NLIM for flagellates and cyanobacteria is higher than 1 during half a year, from November to April. For example, the average (2005-2015) March concentrations of NH4=0.19 and NO23=3.02 uM estimated from monitoring data in the Gotland Deep (BY15) would result in NLIM=0.92 for diatoms but in NLIM=1.13 for others. Consequently, the consistency of interpretations in Section 3.3 must be checked out and corrected as necessary. On the other hand, this entire Section 3.3 should be reconsidered anyway (see below). – In addition to or even instead of Eqs. 14-18 in Section 2.2.2, a comparison between the simulated depth where LTLIM is less than 0.5 (or 0.25, or both) and the mixed layer depth could be more important for the subsequent analysis in Sections 3.4 and 3.5.

2.4.2 Forcing Please, clarify either in the text or in Fig.2's legend: 1) what is shown in Fig.2 – total loads to the entire Baltic Sea or only to the study area as in Fig.1, 2) had the direct point sources been included in the prescribed "river loads" and, thus, assumed seasonally variable as well, 3) see also suggestion to Fig. 2 below.

2.5 Results and discussion 2.5.1 Regardless of whether you'll retain the (very poor and dissatisfying because of variable C:Chl) comparison to observation or will just stay with simulation, the entire Section 3.1 suffers from almost total lack of discussion on the "redistribution" of local maxima, supported by the references to, e.g. Kahru et al., (2016), look also for Wasmund from IOW, Winder, Griffith and their colleagues from Stockholm University, results of AlgaLine, HELCOM, etc.

2.5.2 Nutrient loads. To my mind, you phrasing in the entire Section 3.2 reads as if you imply a casual and almost immediate effect of river inputs at the surface concentrations already at the annual and shorter scale as, for instance, at lines 282-283. Please, consider, at least, two important features of the Baltic Sea: a) long nutrient residence times caused by an order of magnitude difference between residing pools and nutrient amounts annually put into the Sea, b) nutrient exchange of your study area with the south-western Baltic, where the vegetation season starts earlier, and with the northern gulfs with delayed seasonal development. So, never mind the seasonal scales. Perhaps, my further confusion with the longer scales is triggered by the lack of proper explanations on interpretation of "coherence", because I read the entire Section as if it means causal relationship. Adjust your text accordingly to these considerations with appropriate references. A hint – your considerations here could be related to, at least Conley et al. (2002, 2009), Vahtera et al., (2007) and Savchuk (2010). But then the question may arise – how novel are your results and why they are important? Also, what new and important could be in consideration of coherence between river loads and phytoplankton functional types? If nothing substantial, then Figs 6-7 can be painlessly cut off together with corresponding considerations.

2.5.3 Nutrient limitation. Here I have several comments, perhaps, somewhat contradicting one another. – To start with, the entire approach to analysis of simulated variables can be questioned in several aspects, since the analysis and conclusions are based on: a) a specific combination of prescribed constants and could change even with a minor recalibration, b) an inconsistent parameterization of the nitrogen limitation (see above), and c) simulated seasonal dynamics of vertical nutrient distribution that are pretty far away from the observed dynamics (see, e.g. Fig. 5 at p.2120 in Liu et al., 2017). – Furthermore, the implementation of your limiting functions (Eqs. 5-12) instead of a common N:P ratio that directly indicates a deficient/excessive amounts of nutrients gives misleading results. For instance, your finding about persistent winter phosphorus limitation found in the model contradicts, at least, to the situation during recent decades. Note, that for the example given above in 2.4.1, March phosphate concentration was 0.60 uM, which results in DIN:DIP ratio of 5.4., indicating clear N limitation, whereas your Eq. 8 will give PLIM=0.86 for diatoms and PLIM=0.92 for others, which led you to claim the P limitation. – On the other hand, the higher values of N:P ratio (but still well below 16) indicating similar relaxation of the nitrogen limitation relatively to contemporary conditions have also been simulated for the beginning of the XX century, e.g. by Schernewski and Neumann (2005), Savchuk et al., (2008), and Gustafsson et al., (2012). So, I would recommend to repeat your analysis with N:P ratio as more conventional and less questionable indicator of nutrient limitation. – According to Eq. 4, i.e. the minimum law, only P or N must singularly limit at any specific moment. That means that there is simply no place for simultaneous limitation by both P and N together, as shown in Fig. 8. Find, please, less confusing form of presentation, perhaps, with different colors or even contour plots. – Despite my general recommendation about references, the entire paragraph at lines 270-273 does not look as especially necessary here. Instead, in the following considerations at lines 274-280 you would better recall, at least Conley et al. (2002, 2009), Vahtera et al., (2007) and Savchuk (2010). – Consider also, please, moving considerations about coherence between salinity and nutrients from Section 3.2 to Section 3.2 as an explanation of mechanisms transporting results of nutrient redox alterations from deep to surface layers. – On the other hand, there are too many trivial if not erroneous (e.g. occurrence of nitrogen limitation only since the 1980s) and confusing (one periodicity preceding another, how NUTLIM for cyanobacteria accounts for nitrogen fixation) considerations in the entire Section including Figs. 8-17 that should be streamlined and put in the context of existing knowledge as some novel proven findings. – Finally, I would insist on the total re-working of your study on nutrient limitation avoiding dubious and rather inconsistent NUTLIM concept, especially as been applied to seasonally variable mixed

layer and the layer underneath it, down to 150 m.

2.5.4 Perhaps, it would be expedient to put effects of the light limitation and mixed layer depth into some kind of the "critical depth" concept that is studying the period when phytoplankton is not removed from the suitable light conditions for too long

2.5.5 The entire Section 4 must be somewhat shortened and significantly re-written paying attention to: a) avoiding explicit repetitions about simulated results both within Sect 4 and with preceding Sections, b) clearly indicating the temporal scale of every conclusion, c) cardinally reconsidered Section 3.3, especially about unheard-of phosphate limitation of the spring bloom, d) clarification of confusing statements about maxima and phenological terminology (spring, summer, autumn, late summer, etc.).

3. Minor things, technical corrections and language cosmetics Title – Do you need comma after "simulated"? Lines: 12 – Consider, please, replacing "observed" with "found"; 18-19 & 54 – "...functioning of the biology and the biogeochemistry" – vague and slang-like wording, please, find appropriate formulations, preferably helpful in further considerations; 25 – is it "intensification" (hinting at almost immediate response) or rather long-term "accumulation"?; 28-29 – "...the boundary between anoxic and oxic sediments where denitrification occurs also increases." What increases – the length, the area, why only sediments? Consider, please, "the area of interface between oxic and anoxic zones (opt. – i. e. hypoxic zone)"; 36 – consider "rate" or "velocity" instead of "speed"; 64-68 – this mixture of 3D and 1D here is confusing. Since the details as at p. 165 in Eilola et al. (2009) are not necessary here, just simplify appropriately, something like "biogeochemical interactions are described by the SCOBI model"; 87 – "...described IN sections..."; 124 – accordingly to mimicked mechanism, it rather "accounts for inhibition of the nitrate uptake" than "represents preferential ammonium uptake"; 130-131 – Please, explicitly list the order of phytoplankton types in (9-12) 140 – If you think Eq.13 is necessary here, then, please, check the spelling and explain why do you need multiplication of two constant instead of one constant equal to their product; 159-160 – Is the word "matlab" a universally known term, like salinity of nitrate? Is it important to indicate here, or just "... using the algorithms from Jackett et al. (2006)" would do? 223 – "pre-industrial" - something is needed after this adjective: conditions, situation, trophic state, whatever... 245 – which "scarce observations" if you are dealing with the prescribed regular time-series? Do you really use river time series integrated along the entire Baltic Sea coast for studying a coherence with surface nutrients restricted to your study area, regardless of the all shifts and delays in seasonality? 260-261 – why unnecessary "Furthermore, as described above, . . .", where it is important to start from the reminding: "In the model, the effect of nutrients on the primary production. . ." 273 – please, consider carefully ". . .increased deep water respiration but also due to increased temperatures resulting in reduced oxygen solubility." Deep water respiration increases not by itself but due to increased PP and sedimentation. Hypoxic area is delimited by an absolute value of 2 ml/l that has little to do with the relative solubility. 280 – ". . .increased riverine loads. . ." Even according to your Fig.2, they have been decreasing since the 1980s! I think, "the accumulated terrestrial inputs" would be better description; 282-283 – here you go again ". . .since the mixed layer is directly affected by riverine input. . ." On which scale?! Please, estimate pools and compare to inputs! 283 – "The mixed layer also comprises a smaller volume of water." So, what? Besides, 4,000 (volume of layer 0-27 m) out 12,000 km3 doesn't look small to me. Fig. 2 – Here and everywhere with the curves I recommend using pretty common convention about color and chemical elements (https://en.wikipedia.org/wiki/CPK_coloring), at least, strictly using blue for N and then red or purple for P. Besides, it could be more logical showing all loads on one graph and concentrations on another. Fig. 4 – why different definitions – is "surface DIN concentration" also a mixed layer concentration similar to phosphate?

---

## Author Comment (AC1) · 2 Aug 2017

**Response to referee O.P. Savchuk**

We thank the referee for his extensive and insightful comments that will greatly improve the manuscript. Our responses are listed below.

1.1 As we understand it, the referee states that causes are already implicitly known since we know the governing equations together with boundary- and initial conditions and that what is needed is just a clever diagnostic of the fluxes. We disagree. In fact, we do not know of a single solution to the Navier-Stokes equations that govern the fluid dynamics of the problem. A consequence of this is that

[Figure]

there is not one single dynamical theory which is founded on the full equations of motion. It is thus by no means, in general, simple to tie some observable quantity, such as a chlorophyll, to a boundary condition. This is, in turn, why we use language such as small scale mixing caused by breaking internal waves even though breaking internal waves are solutions to the equations of motion rather than a boundary condition. The usage of the word "causes" in this way is very common, and we don't see anything wrong with it.

Furthermore, we are aware that this is not the only study of its kind. However, the coherence of different variables on inter-annual time-scales, as considered here, have to our knowledge not been scrutinized before in the Baltic. The referee suggests that our approach needs a specific motivation and justification. However, we look at co-variations between different variables on inter-annual time-scales, which is a very much a classical line of inquiry in many scientific disciplines, but using a relatively novel method and do not see what further justification that could be needed.

1.2 We agree and intend to remove the observational part as is suggested by the referee in a later point. We will also improve the methods section to include a more extensive description of wavelet coherence. Indeed, we will clarify the text so that it is clear that we do not mean to imply causality.

1.3 We agree and will change this according to the suggestions.

1.4 Our main focus is on longer time-scales. However, we have chosen to include seasonality where we see a shift from an earlier different pattern such as during the mid 1970s. We will review and remove the seasonality where we find that we can.

1.5 We will go with the suggestion and remove the observational part. The big problem with observations is that the datasets required for inter-annual coherences

do not exist. Long time-series exist only in some spots, and even there, there are typically gaps in the data that makes them unfit for wavelet analysis. This is of course a problem that cannot be helped through literature references.

1.6 We agree with the reviewer and will improve the text and structuring to better highlight the Why?, How? and What?. There are not many references to works on the inter-annual time-scale in the manuscript because few exist. The lack of observations was discussed above.

2.1 A title is supposed to be both informative and to some extent catchy. I think very few people will be interested in having a very intricate division of the Baltic Sea in a title. We will change the title to: "Causes of simulated long-term changes in phytoplankton biomass in the Baltic Proper: a wavelet analysis". The usage of the word "causes" is discussed already in 1.1.

2.2 We will change internal loads to sedimentary release.

2.3 We will improve the introduction by emphasizing the questions and knowledge gap for inter-annual time-scales but we think that the justification for using wavelets is obvious. Classical spectral analysis cannot be used to study temporal changes in frequency, while wavelets can. That is why we use them.

2.4 This will be corrected.

2.4.1 We understand the objections and will rethink what equations we need to include. Indeed, the referee is right and the statement in lines 102-103 about the relation between salinity and the sedimentary phosphorus release is wrong. However, the statements in lines 251-253 do not mean to imply that an increase in sedimentary phosphorus release due to high salinity has anything to do with the observed pattern. Rather, salinity is here thought of as a proxy for water exchange. If mixed layer salinity shows in-phase coherence with mixed layer phosphate while

DIN shows anti-phase, it indicates that the phosphate is accompanied by high salinity water (perhaps upwelled from deeper layers) while DIN is accompanied by low salinity water. We will clarify this and search for any further problems relating to the mistake on lines 102-103. The questions about nutrient limitations are discussed in our answer to 2.5.3.

2.4.2 It is only around the study area. This will be clarified in the updated version.

2.5.1 Section 3.1 will be removed.

2.5.2 Riverine nutrient input is directly put into the mixed layer and the effect is obviously immediate in the vicinity of the river mouth. Further out it is likely set by an advective time-scale (L/V) for the Baltic Proper. This is not to say that exchange with other basins is not important (the reviewers point b)), but we have not quantified those exchanges in this manuscript.

Long residence times for nutrients in the Baltic are in point a) suggested to be a reason for why riverine input should be unimportant for annual and shorter time-scale biomass fluctuations. This is clearly incorrect. The mixed layer nutrient pool is depleted every year. And the fact that the residence time for nutrients in the Baltic as a whole is long appears to us to have very little to do with short time scale biomass fluctuations.

To conclude, we are not convinced by the referees suggestion that riverine input is unimportant on short time-scales. A clear coherence between riverine input and mixed layer concentrations on an annual time-scale in the wavelet analysis is also evident. This, of course, does not imply a causal relationship, but it is to our mind clearly worthy of note. About the use of the term coherence; it is standard mathematical lingo and it does not imply a causal relationship. Surely, this is not something that needs explaining in a science paper.

2.5.3 a) and c) All climate models are sensitive to parameter choices and analysis of

simulated variables are standard scientific practise. We can't see any reason why these practices need defending here.

b) It is true that NLIM under certain conditions may obtain values >1 but this does not imply that this would amplify growth. PLIM <1 gives P limiting conditions if NLIM >1. NUTLIM is the quantity that the model cares about and it is always 0 <NUTLIM <1. No problem there apart from the cosmetics of having NLIM >1. This answers also the questions put forth in 2.4.1 about the limiting functions.

Regarding limiting functions. We are analysing modelling results and thus we need to use the rules that govern the models growth rate. Therefore, we disagree that it would be better to use N:P-ratios and believe that using some other limitation than what determines the growth in the model would be very confusing. Thus, the statement that the usage of the NUTLIM concept in our analysis leads to misleading results is incorrect, as it illustrates the actual workings of the model. However, we think that the criticism put forth by the reviewer that NUTLIM and N:P ratios may lead to different nutrient limitations is important, and we will highlight this in the revised manuscript. About fig. 8 it does not show simultaneous limitation by both nutrients. Rather it is the size of the rings that indicate the monthly values that give this appearance, we will add a note in the figure caption.

2.5.4 For some quantities this might be helpful but we prefer the mixed layer concept, as is much more straight forward when it comes to the physical quantities. The sharp pycnocline inhibits vertical transfer, and is therefore a more natural choice for studying variations in N and P concentrations

2.5.5 We agree with the referee and will rewrite section 3.3 in accordance with the review comments. The phosphate limitation of the spring bloom produced by the model will be discussed. This can also contribute to the new discussion about the NUTLIM vs N:P ratio concept that will be added

3. We have no objections to the minor comments, with the exception of the riverine

input where we have given our view in 2.5.2

---

## Author Comment (AC2) · 2 Aug 2017

**Response to Referee #3**

We thank the referee for the valuable comments that will significantly improve the manuscript. Answers to comments are listed below. Referee comments in italics.

Reworking and shortening section 2.2 has been suggested by both referees. We agree and will review this section accordingly.

- *Already in the abstract combination of words "mixed layer 'parameter' concentrations" appears as solid term. However I did not find in the text how it was defined.*

[Figure]

*Is it mean value of horizontal mean 'parameter' in horizontal mean mixed layer? or it is integrated characteristic?*

The definition of mixed layer concentrations is

$$\frac{\int_{ML} C dV}{\int_{ML} dV}$$

. We will clarify in the manuscript.

- *Salinity in the Baltic Sea and in the Baltic Proper have strong lateral gradient. However, mixed layer depth (MLD) was defined as constant density difference. Could it be that with decrease of salinity MLD will increase? Could it be that seasonal variability in surface effects MLD and at the end all results ? The part with mixed layer definition should be extended and some how emphasized. Maybe it makes sense to include it as additional subsection.*

Horizontal gradients in S should not matter but weaker stratification might. However, we have carefully checked that it appears to find the pycnocline everywhere in the domain. The problem might have been bigger if the whole Baltic had been used, rather than just the Baltic proper though.

- *The "basin integrated approach" was used here (line 61). Would be good to see in the text why this is acceptable (preferably in more than one sentence, line 6*

This approach is basically more robust than a point value, as it takes away some small scale variability that we are not really interested in here. A 1d dataset is also required for the wavelet analysis.

- *While SCOBI model is 1D model (line 67), I would suggest to show results of wavelet analysis for idealized 1D cases. So it could be seen how certain changes are reflected in final results of wavelet analysis. For my opinion such sensitivity*

*test could enhance conclusions. Otherwise, section 2.4 should be extended with some aspects of wavelet coherence.*

All our wavelets are 1d. As stated above, the wavelet approach uses a 1d dataset. The only dimension we have is time. The spatial dimensions are removed by taking spatial averages.

- *Analysis focuses mainly on river loads and its changes. Other nutrient sources like atmospheric deposition, exchange with other Baltic Sea regions and there possible effect should be mention somehow*

This is a good point. We will elaborate more on this in the revised version

- *It could be considered to include wavelet analysis in to the title – to my opinion application of this method is among the most interesting aspects of this manuscript*

We will add this to the title.

- *Line 75: eq. 1. NFIX is nitrogen fixation term, in all phytoplankton groups it looks strange. Is it a misprint?*

NFIX is zero in non-nitrogen fixing phytoplankton groups. We will put a note in the revised manuscript.

- *Line 78: SINKIphy / SINKOphy is it sinking of phytoplankton?*

Yes, in/out of the gridbox.

- *Lines 177 - 181: Paragraph is confusing. It starts with sentence about open boundary, but last two sentences are probably about river loads. Please specify in more details: what these assumption were applied to*

We will rewrite.

---

## Author Response (AR1)

**Response to referee O.P. Savchuk**

We again thank the referee for his valuable comments on the manuscript. In addition to our previous response we will here provide som further comments on the changes that have been made in accordance with the referee feedback.

1.1 We have removed implications of causality where possible. We do however retain the word "causes" in the title with reference to our earlier reply.

1.2 We have removed the observational part. We have instead chosen to present the time-series and the wavelet spectrum of the simulated phytoplankton biomass together with the month of maximum chlorophyll maxima in Sect. 3.1. We have tried to make it clear that we are using only simulated variables.

1.3 We have changed to "phytoplankton biomass" as well as added a comment on the constant C:Chl ratio in Sect. 2.2.

1.4 We have removed discussions around the seasonal time-scale where possible. We have kept comments on the seasonal scale for the clear regime shift shown in current Fig. 10.

1.5 Again, we have removed the observational parts from the manuscript.

1.6 We have tried to improve the structuring and motivation for the manuscript mainly in the introduction and throughout Sect. 3. Much of the justification certainly boils down to the use of a relatively new tool. However, as a similar analysis has not previously been done for simulated biogeochemical variables we feel that an illustration of its uses is valuable.

2.1 We have changed the title to: Causes of simulated long-term changes in phytoplankton biomass in the Baltic Proper: A wavelet analysis.

2.2 We have removed "internal loads" where possible.

2.3 As stated in 1.6, we have tried to rework the introduction and section 3 in accordance with the referee comments.

2.4 This has been corrected.

2.4.1 We have removed unimportant equations from section 2. We have also removed the faulty comment on phosphate and salinity.

2.4.2 We have clarified that it is only around the study area.

2.5.1 The comparison with observations have been removed

2.5.2 We have removed discussion on the seasonal scale for the riverine input. We have further rewritten the section so that it is clear that we do not imply causality. We have also removed previous Figs 6 and 7.

[Figure]

Figure 1: Nitrogen or phosphate limitation as calculated with N/P ratios.

2.5.3 We have added a comment on that Nlim can become larger than one
(but not NUTLIM). However, as NUTLIM is what directly affects the
phytoplankton growth in the model we have kept this formulation. We
have added a discussion on that N/P ratios gives a different result more
inline with observations in Sect. 3.2 (see Fig. 1).

We have also added a note in the figure caption that simultaneous N and
P limitation is not possible.

2.5.4 For some quantities this might be helpful but we prefer the mixed layer
concept, as is much more straight forward when it comes to the physical
quantities. The sharp pycnocline inhibits vertical transfer, and is therefore
a more natural choice for studying variations in N and P concentrations.

2.5.5 We have tried to rework the section (now 3.2) so that the purpose of the
section is more clear. We have kept the figure showing our model results
for anoxic volume and deep water nutrients (now Fig. 5) since we believe
it to be necessary for the discussion.

3. We have adressed the minor comments.

**Response to Referee #3**

We again thank the referee for the valuable comments. Referee comments in italics.

We have shortened section 2.2 and removed unnecessary equations.

- *Already in the abstract combination of words mixed layer parameter concentrations appears as solid term. However I did not find in the text how it was defined. Is it mean value of horizontal mean parameter in horizontal mean mixed layer? or it is integrated characteristic?*

  We have clarified that the parameters are horizontally and depth *averaged* and not integrated.

- *Salinity in the Baltic Sea and in the Baltic Proper have strong lateral gradient. However, mixed layer depth (MLD) was defined as constant density difference. Could it be that with decrease of salinity MLD will increase? Could it be that seasonal variability in surface effects MLD and at the end all results ? The part with mixed layer definition should be extended and some how emphasized. Maybe it makes sense to include it as additional subsection.*

  Changes in salinity will effect the mixed layer depth due to its effect on density, but this is captured in the definition of mixed layer as long as the equation of state used to calulate the density difference depends on salinity. The definition also captures the seasonal changes in mixed layer depth when monthly mean profiles of salinity and temperature are used in the calculation. The mixed layer definition is also standard and frequently used both in models and on observational data.

- *The basin integrated approach was used here (line 61). Would be good to see in the text why this is acceptable (preferably in more than one sentence, line 6*

  We have added a comment in section 2.1.

- *While SCOBI model is 1D model (line 67), I would suggest to show results of wavelet analysis for idealized 1D cases. So it could be seen how certain changes are reflected in final results of wavelet analysis. For my opinion such sensitivity test could enhance conclusions. Otherwise, section 2.4 should be extended with some aspects of wavelet coherence.*

  We have improved section 2.4 to provide a better description of the wavelet transform and wavelet coherence.

- *Analysis focuses mainly on river loads and its changes. Other nutrient sources like atmospheric deposition, exchange with other Baltic Sea regions and there possible effect should be mention somehow*

  We have tried during the revision work to include the atmospheric deposistion. Sadly, we only had yearly averaged values of the deposition to work with and that is not good enough for the wavelet analysis. The horizontal transports suffers from a similar problem. Here we have the velocity fields and concentration fields but not their products, and we thus do not really know the transports. In future work we plan to close these nutrient budgets using online calculations, but in this current effort we have settled to look at correlations with some of the most important forcings.

- *It could be considered to include wavelet analysis in to the title to my opinion application of this method is among the most interesting aspects of this manuscript*

  We have changed the title in accordance with the review comment.

- *Line 75: eq. 1. NFIX is nitrogen fixation term, in all phytoplankton groups it looks strange. Is it a misprint?*

  We found the equation to be unnecessary and have therefore removed it.

- *Line 78: SINKIphy / SINKOphy is it sinking of phytoplankton?*

  We have removed the equation.

- *Lines 177 - 181: Paragraph is confusing. It starts with sentence about open boundary, but last two sentences are probably about river loads. Please specify in more details: what these assumption were applied to*

  We have rewritten.

[revised manuscript text omitted]

$\text{S}_{\text{PHY}} \quad = \quad \text{GROWTH}_{\text{PHY}} + \text{NFIX} + \text{SINKI}_{\text{PHY}}$

$\qquad\qquad\qquad -\text{SINKO}_{\text{PHY}} - \text{MORT}_{\text{PHY}} - \text{GRAZE}_{\text{PHY}},$

phytoplankton biomass is described in terms of chlorophyll and with a constant C:Chl ratio. The model thus does not take into account seasonal changes in C:Chl as was found by Jakobsen and Markager (2016) .

The net growth of phytoplankton is described by the following expression,

$\text{GROWTH}_{\text{PHY}} \quad = \quad \text{ANOX} \cdot \text{LTLIM} \cdot \text{NUTLIM}_{\text{PHY}} \cdot \text{GMAX}_{\text{PHY}} \cdot \text{PHY},$ (1)

where subscript PHY indicates the plankton funktional type (diatoms, flagellates or cyanobacteria). ANOX is a logarithmic expression that approaches zero as the oxygen concentration becomes small.

LTLIM expresses the phytoplankton light limitation and NUTLIM describes the nutrient limitation. Nutrient limitation follows Michaelis-Menten kinetics where constant Redfield ratios are assumed in nutrient uptake. NUTLIM  is further described in Sects. 2.2.1 and 2.2.2. GMAX is temperature dependent and describes the maximum phytoplankton growth rate.

The difference between diatoms and flagellates are present in halfsaturation constants, maximum growth rate, temperature dependence and sinking rate. Flagellates are more sensitive to a change in temperature than diatoms. Furthermore, the sinking rate of diatoms is five times larger than that for flagellates.

The difference between cyanobacteria and the other phytoplankton species is more pronounced. Cyanobacteria can grow either according to Eq. (1) or using nitrogen fixation

$\text{NFIX} \quad = \quad \text{ANOX} \cdot \text{NF} \cdot \text{A3}$

. The rate of nitrogen fixation as a function of the phosphate concentration and temperature,

. Both NFIX and GROWTH of cyanobacteria is zero if the salinity is above 10. Furthermore, cyanobacteria is the most temperature sensitive of the phytoplankton groups and no sinking velocity is assumed.

Other processes important for our results involves chemical reactions occurring in the water column or in the sediment.

Denitrification occurs in both the water column and the benthic layer and constitutes a sink for nitrate in case of anoxia.

Nitrification transforms ammonium into nitrate as long as oxygen is present. Phosphorus is adsorbed to the sediment and the benthic release capacity of phosphate is a function of the oxygen concentration where more oxygen implies less release. The phosphorus release capacity is also dependent on salinity where higher salinity means  less phosphate is retained in the
benthic layer.

**2.2.1 Nutrient limitation**

Estimating nutrient limitation in nature is difficult. Usually this is done, either by comparing nutrient ratios to Redfield in eg.
the surface water or external supply or by some nutrient enrichment experiment (Granéli et al., 1990).

The idea of nutrient limitation as often used is based on that the primary production is directly limited by the nutrient
concentration in the ambient water and that the internal nutrient ratios in the phytoplankton are constant, i.e. in accordance with
a Redfield-Monod model (Redfield, 1958). However, cell-quota type models (Droop, 1973) are being increasingly implemented
and the use of constant internal nutrient ratios are becoming more and more questioned (Flynn, 2010).

Furthermore, N vs P limitation is a long standing debate. Tyrrell (1999) uses a box-modelling approach to show that in
steady state, nitrogen becomes slightly deficient while it is the external input and removal of phosphate that ultimately controls
the production.

Here, nutrient limitation is traditionally expressed assuming constand Redfield ratios and phytoplankton growth is limited
by either nitrogen or phosphate. The degree of nutrient limitation is described by:

$$\text{NUTLIM}_{\text{PHY}} \quad = \quad \min(\text{NLIM}_{\text{PHY}}, \text{PLIM}_{\text{PHY}}) \tag{2}$$

where $\text{NLIM}_{\text{PHY}}$ and $\text{PLIM}_{\text{PHY}}$ are the nitrogen and phosphate limitation respectively. In addition, $\text{NLIM}_{\text{PHY}}$ contains the
sum of the nitrate and ammonium limitation, i.e.

$$\text{NLIM}_{\text{PHY}} \quad = \quad \text{NO}_3\text{LIM}_{\text{PHY}} + \text{NH}_4\text{LIM}_{\text{PHY}}, \tag{3}$$

where

$$\text{NO}_3\text{LIM} \quad = \quad \frac{\text{NO3}}{\text{KNO3}_{\text{PHY}} + \text{NO3}} \cdot \exp(-\phi_{\text{PHY}} \cdot \text{NH}_4), \tag{4}$$

$$\text{NH}_4\text{LIM} \quad = \quad \frac{\text{NH4}}{\text{KNH4}_{\text{PHY}} + \text{NH4}}, \tag{5}$$

where NO3 and NH4 are the concentrations of nitrate and ammonium and $\text{KNO3}_{\text{PHY}}$ and $\text{KNH4}_{\text{PHY}}$ are the halfsaturation
constants for nitrate and ammonium respectively. The exponent in (4)  accounts for inhibition
of nitrate uptake (eg. Dortch (1990); Parker (1993)).

$\text{PLIM}_{\text{PHY}}$ is modelled as,

$$\text{PO}_4\text{LIM} \quad = \quad \frac{\text{PO4}}{\text{KPO4}_{\text{PHY}} + \text{PO4}}. \tag{6}$$

Nutrient limitation is thus described by a number between 0 and 1 where 1 is no limitation. Note that NLIM in Eq. (3) may
obtain values larger than 1. However, as NUTLIM is calculated as the minimum of NLIM and PLIM, NLIM larger than one
will always mean P limitation.

The constants $KNO3_{PHY}$, $KNH4_{PHY}$ and $KPO4_{PHY}$ are the half saturation constants and differs between the different
phytoplankton groups. The constant $\phi_{PHY}$ in Eq. (4) determines the strength of ammonium inhibition of nitrate uptake. The
values of the constants for each phytoplankton type are given below.

$KNO3_{PHY} \equiv 0.5/0.25/0.25$

$KNH4_{PHY} \equiv 0.5/0.25/0.25$

$KPO4_{PHY} \equiv 0.1/0.05/0.05$

$\phi_{PHY} \equiv 1.5/1.5/1.5$

Note that the half-saturation constants for flagellates and cyanobacteria are equal which means that in absence of nitrogen
fixation, the nutrient limitation for the nitrogen fixing species is equal to that of flagellates.

In addition to the above given nutrient limitation of phytoplankton growth there exists a similar nutrient dependency on
nitrogen fixation. In the model this dependency reads

$$NUTLIM_{NF} \equiv \frac{aNFC}{aNFC + \left(\frac{NO_3 + NH_4}{PO_4 cNFC} dNFC\right)}) \cdot \frac{PO_4}{\alpha NF \cdot \beta NF + PO_4},$$

where aNFC, bNCF, cNFC and dNFC are constants used for calculating the nitrogen fixation capacity which in turn is a
function of the ratio of inorganic nitrogen to phosphate. $\alpha NF$ and $\beta NF$ are constants determining the half-saturation for nitrogen
fixation. Again, $NUTLIM_{NF}$ approaches one if the nitrate and ammonium concentrations are zero and for large concentrations
of phosphate.

**2.2.2 Effect of physical parameters**

Changes in cloud-cover affect the incoming solar radiation and thereby the phytoplankton growth. The effect of light shows up
in the LTLIM term of Eq. (1).

$$\quad \underline{\mathrm{LTLIM}} \quad \equiv \quad \underline{\mathrm{fracI_{PAR}I_{opt} \cdot EXP\left(1 - \frac{I_{PAR}}{I_{opt}}\right)},$$

$$\quad \underline{I_{PAR}(z)} \quad \equiv \quad \underline{\alpha_{PAR}I_0 \cdot EXP\left(-Kd \cdot z\right)}$$

$$\quad \underline{I_{opt}} \quad \equiv \quad \underline{\max(I_{opt,min}, \alpha_{opt}I_0)}$$

$$\quad \underline{Kd} \quad \equiv \quad \underline{Kd_w + Kd_{PHY} + Kd_Y + Kd_D}$$

$$\quad \underline{Kd_{PHY}} \quad \equiv \quad \underline{\alpha_{Kd}(A1 + A2 + A3)}$$

[revised manuscript text omitted]

---

## Author Response (AR2)

**Response to referee O.P. Savchuk**

We thank the referee for his rigorous and insightful comments on the manuscript. Our responses are listed below. Referee comments in italics.

**Major concerns**

1. *In the revised version, it became even clearer that the manuscript deals with a study of the model itself rather than with a usage of the model for studying the marine system. As the authors admitted: Much of the JUSTIFACATION certainly boils down to the USE of a relatively NEW TOOL. we feel that an ILLUSTRATION OF ITS USES is valuable. As deficit of geographical and biogeochemical plausibility cannot be compensated by methodological novelty of the implemented statistical tool (wavelet analysis), such study is more appropriate, perhaps, for the journals dealing with modelling technics rather than in Biogeosciences (see my initial comments).*

   We understand the concerns put forward by both referees regarding the lack of comparison with real data. We have added references to studies that includes validation of nutrients, temperature and salinity for the particular model run used in our study. Validation of the different plankton species is difficult due to lack of observations and essentially unknown C:Chl ratio, and had not previously been done for the model run we use. We have now added such an evaluation of the biology, assuming a fixed C:Chl ratio of 50 shown in Fig. 3 and 4. The fixed ratio is likely responsible for part of the model-observation difference in absolute values, but has probably a negligable impact on the timing of blooms.

2. *The revisions concerning nutrient limitation are still confusing and misleading in respect to the real Baltic as we know it from both observations and simulations with other models. No consideration is given to how an appropriate re-parameterization of internally inconsistent N-limitation ¿ 1 would change relations between NLIM and PLIM for different phytoplankton groups. More importantly, the chosen method compares N- and P-limitation as determined by a specific set of constants prescribed in result of calibration of this specific model, while a sensitivity of simulation to such a choice is not even mentioned. The manuscript still contains a little addition in- , perhaps, even directly contradicts to existing knowledge about the relationships between nutrient and phytoplankton dynamics in the Baltic Proper. In that respect, particularly surprisingly and unconvincingly sound such expressions as a shift towards less limited conditions (Line223), Phosphate is still limiting during winter (L230), the phase shifts from NUTLIM preceding diatoms by three months to diatoms preceding nutlim by the same amount (L245-246) and the following considerations of maximal NUTLIM on Fig.11 (shouldnt minimum NUTLIM be more interesting?), The spring bloom is phosphate limited throughout*

*the run except for a few years after 1990 where diatoms display nitrogen limitation (L304-305). Correspondingly, the entire Section 3.2, including Figs. 6-11 still looks just as exercise with a new tool, being more concerned with the tool rather than the Baltic. Perhaps, it can be focused only on analysis of N:P instead of NUTLIM and substantially shortened, absorbing and compressing much from the Summary and Conclusions Section as well.*

As stated in our answers to the previous review comments, if NLIM is larger than one, PLIM is used. NUTLIM, which is what is used in the model, never becomes larger than one. To clarify, we have introduced a condition statement in Eq. 3 that states that NLIM can not be larger than one. Furthermore, we have included a discussion in Section 2.1.1. (lines 107-111) on that the nutrient limitation is sensitive to the choice of parametrization.

We have also added a discussion around reconstructed limitation patterns compared to the models limitation pattern using N:P ratios and the models inherent definition, NUTLIM, in section 3.2.

The limitation patterns calculated by NUTLIM and N:P ratios clearly show the same trend from P to N limitation over the 20th century but vary quite a bit for different months (Fig. 9). Which of these limitation functions best capture the real world is hard to say, given how variable the Redfield ratio has been found to be in phytoplankton. However, for our model results this is not a problem, as these are undeniably controlled by NUTLIM.

3. *The laconic Section on river loads looks now better. Although, the strong inter-annual coherence (with only 1 year lag?) between local riverine input and DIN in the mixed layer (L255) deserves more consideration and explanation, remembering although about open boundaries with the gulfs and the south-western Baltic. On the other hand, the anti-phase coherence between salinity and DIN on periodicities ¿1 yr, might has the same reason as the in-phase coherence between salinity and phosphate, that is the upward transport of deep waters due to hypoxia enriched with DIP and depleted in DIN. Could be worth mentioning as pertinent to the vicious circle of Baltic Sea eutrophication that is more appropriate here than in the Section on limitation.*

We have added a discussion on this in section 3.3.

4. *The Summary and conclusion Section contains too much repetition or just a prolongation of the awkward discussion from the preceding Results and discussion Section, including even some contradictions with dates and chronology. It must be effectively cut down to just the conclusions.*

We agree and have significantely shortened section 4. focusing on just the conclusions.

**Minor suggestions**

We have revised the manuscript in accordance with the minor suggestions.

**Response to Referee #3**

We thank the referee for valuable comments on the manuscript.

> The concern about lack of comparison with observations were put forward by both referees. We have added references to studies which includes validation of the particular run used in our manuscript. We have further included a validation of simulated phytoplankton at monitoring station BY15.

> The minor comments have been adressed.

[revised manuscript text omitted]

---

## Author Response (AR3)

**Response to associate editor**

A great thanks to the associate editor for the rigorous work put into improving the manuscript. Below are our replies to the editors comments (in bold font).

**below are a few comments that should be considered before submitting the final version for publication. While I understand that the method used requires long time series, it would be helpful to add some information (a few sentences) on how the results from the simulation and the analysis presented in the manuscript could be used to improve interpretation of field data, or alternatively be validated by field observations. Further, the comparison with field data on plankton groups (i.e. Fig. 3) could be more detailed, in particular, a discussion on whether the mismatch (for example cyanobacteria seasonal dynamics and biomass after 2005) is due to differences between field and model environmental parameters presented in the analyses (nutrients, light, mixed-layer depth, temperature and salinity) or to the representation (growth parameters) of the different plankton groups in the model? This would make a valuable addition to the manuscript.**

At present, we have focused on the validation by field observations. This section has been lengthened with more detailed discussions.

**Equation 1 (lines 78 and below): while the meaning of the subscript PHY is explained, no information is given about the parameter PHY at the end of the equation.**

We have revised the manuscript in accordance with your comment.

**p. 5, lines 120-122: could be replaced with "...half-saturation constants for nitrate and ammonium uptake, respectively. The exponent in (4) accounts for inhibition of nitrate uptake in the presence of ammonium (e.g. Dortch, 1990; Parker, 1993)".**

We have revised the manuscript in accordance with your comment.

**p. 5, line 122 and equation 6: Why introduce a new term "PO4LIM" when this parameter is the same as PLIMPHY already in use? The author could simply use PLIMPHY in eq. 6.**

We have revised the manuscript in accordance with your comment.

**p. 5, lines 124-131: Reorganize and shorten paragraph (redundant information) as given below:"The constant KPO4PHY is the half saturation constants for phosphate. Nutrient limitation NUTLIM is thus described by a number between 0 and**

1, where 1 corresponds to no limitation. Since NUTLIM is calculated as the minimum of NLIM and PLIM, NLIM larger than PLIM will temporally cause P limitation of phytoplankton growth rate. Hence, a different formulation e.g. of NLIM might change a models sensitivity to the limiting nutrient. Its impact on system nutrient dynamics on longer time scales is, however, difficult to judge because e.g. nitrogen fixation and denitrification potentially also may be influenced. Further experiments on this issue are out of the scope of the present paper and left for future studies."

We have revised the manuscript in accordance with your comment.

**p. 5, lines 129-130: "Its impact on system nutrient dynamics on longer time scales is, however, difficult to judge because e.g. nitrogen fixation and denitrification potentially also may be influenced". How so? Can the authors be more specific? Or remove this sentence? Since the model seems to be a standard model for the region, aren't there any sensitivity studies using SCOBI available that can be cited from the literature?**

Sensitivity studies with SCOBI using different NUTLIM expressions have not been performed in any published material. Nitrogen fixation and denitrification will be impacted since the cyanobacteria is affected by the NUTLIM expression. Therefore, the nutrient composition and thereby denitrification will probably also be affected. We have chosen to follow your suggestion and have removed the sentence.

**p. 6 line 154: can be replaced with " the nutrient inputs from rivers and point sources between 1970 and 2006 were compiled from..."**

We have revised the manuscript in accordance with your comment.

**p. 6 line 157: can be replaced with " Atmospheric inputs were estimated in a similar manner based on data from Ruoho-Airola et al., 2012."**

We have revised the manuscript in accordance with your comment.

**p. 6 lines 158-159: can be replaced with "For riverine organic phosphorus and nitrogen inputs a bioavailable fraction of 100% and 30%, respectively, was assumed."**

We have revised the manuscript in accordance with this and your following comment.

**p. 6 lines 158-159: What about atmospheric organic N and P inputs?**

Atmospheric loads include organic N only. We have added this to lines 135-136 in the manuscript.

**p. 6 lines 171-172: can be replaced with "Organic nitrogen was implicitly added through the Redfield ratio (nitrogen to phosphorus) of detritus in the model".**

We have revised the manuscript in accordance with your comment.

**p. 7, line 177: can be replaced with "The phytoplankton functional groups in the simulations and respective observations from station BY15 are shown in Figure 3 and 4."**

We have revised the manuscript in accordance with your comment.

**p. 7, lines 178-181: can be replaced with "Phytoplankton biomass from field observations has been estimated through the conversion of biovolumes into carbon in accordance with Menden-Deuer and Lessard (2000). Phytoplankton biomass for the model simulation was estimated from chlorophyll (Chl) assuming a C:Chl ratio of 50. This ratio is in the middle of the salinity dependent range found by Rakko and Seppl (2014)."**

We have revised the manuscript in accordance with your comment.

**Section 2.4 (Evaluation): This section is an interesting and important addition to the manuscript but it could be transferred to results section 3.1. The comparison indicates some aspects that are not mentioned and unlikely to be simply due to the conversion factor as hypothesized in lines (188-190). These are: - The model does not properly predict diatom blooms. - The model significantly overestimates Cyanos from 2005 onwards while nanoflagellates are overestimated between 1999-2001.**

There are many things that influences the accuracy of the model results of phytoplankton biomass. Specifically, the phytoplankton abundance and distribution depend on the nutrient concentrations, DIN and DIP, and the relationship between them. The errors introduced in nutrients due to e.g. errors in horizontal or vertical transport transfers to the phytoplankton. Furthermore, the frequency of plankton observations is much lower (once a month) than the model output frequency (two-daily) making it much more likely to miss peak biomass in the observed dataset. The observations can also be affected by circumstances such as patchiness during in-situ sampling.

Regarding the specific over- and underestimations shown in Fig 3; winter nutrient concentrations 2006 and 2007 at monitoring station BY15 are too high which explains the strong diatom blooms during these years. Furthermore, the relationship between modelled N and P differ from reality which introduces errors in the distribution of plankton functional types. This may in part explain the overestimation of diatoms and underestimation of flagellates and others during 1999 and 2000.

The simulated cyanobacteria blooms occur too late in the year. On-going work with implementing a cyanobacteria life cycle model shows significant improvements especially in the timing of blooms.

We have added a discussion around this in section 2.4.

We have decided to keep the section where it is as opposed to moving it to Results. This is because we do not think of the model evaluation as a result of the study. The results also deals with horizontally and vertically averaged model data compared to the evaluation which presents data from one monitoring station.

**p. 7, lines 192-194: can be replaced with "The wavelet transform method and its applications have been described in several studies (e.g. Lau and Weng, 1995; Torrence and Compo, 1998; Grinsted et al., 2004; Carey et al., 2016). Below we provide, therefore, only a brief overview of the method."**

We have revised the manuscript in accordance with your comment.

**p. 8, lines 215-218: can be replaced with "The disadvantage of wavelet transform analysis is that it requires long datasets without gaps, while on the temporal scale of climate change such observations on plankton dynamics are lacking."**

We have revised the manuscript in accordance with your comment.

**p. 8, lines 238-239: can be replaced with "...the spring and autumn blooms, further, the power of both periodicities increases markedly after 1950."**

We have revised the manuscript in accordance with your comment.

**Further, the authors use the word "visable" in line 238 and in other parts of the manuscript. Replace visable (not an english word) with visible or synonyms (i.e. noticeable, discernible etc...).**

We have revised the manuscript in accordance with your comment.

**p. 9, lines 243-245: Fig. 3 indicates that cyanobacteria biomass in the model is overestimated from 2003 onwards. Isn't that the reason the Chl maxima shifts towards August-September?**

Good point, but the shift is visible also in 1999 and 2001 when the cyanobacteria biomass at BY15 is not significantly overestimated. It is also difficult to draw conclusions of cyanobacteria biomass for the entire Baltic Proper from only one monitoring station. The model generates a persistent increase in cyanobacteria biomass from the 1970s. The reason is that cyanobacteria biomass increases more than the biomass of the other phytoplankton groups. As stated in the manuscript Kahru et al. (2016) found a similar shift using satellite observations.

**p. 9, line 248: replace "a strengthening of primary production" with "an increase in primary production"**

We have revised the manuscript in accordance with your comment.

**p. 9, lines 251-252: replace with "This has led to a change in nutrient availability and dynamics as anoxia leads to a release in sedimentary phosphate (Conley et al., 2002; Savchuk, 2010)**

We have revised the manuscript in accordance with your comment.

**Legend Figure 7: why is ammonium included in the term DIN? Replace with "Time-series of volume of anoxic waters (top panel), deep water concentrations of nitrate + ammonium (blue) and phosphate (red) (middle panel) as well as nitrate (blue) and ammonium (red) (bottom panel). Deep water concentrations where averaged below the mixed layer depth for the Baltic proper."**

We have used the standard definition of Dissolved Inorganic Nitrogen (DIN) as the sum of the inorganic nitrogen species (e.g. HELCOM, 2017). For the model, which only includes nitrate (model nitrate can be viewed as the sum of nitrate and nitrite) and ammonium, DIN is thus

$$DIN = NO_3 + NH_4^+$$

.

If we have missed some new development on how DIN should be defined, please provide a reference.

HELCOM (2017). Dissolved inorganic nitrogen (DIN). HELCOM core indicator report. Online. Viewed 2018-06-25. http://www.helcom.fi/baltic-sea-trends/indicators/dissolved-inorganic-nitrogen-(din)/assessment-protocol/

**p. 9, line 271: replace with "...Fig. 8. Mixed layer values of NUTLIM increase over the 20th century indicating higher nutrient load and less nutrient limiting conditions."**

We have changed to: Mixed layer values of NUTLIM increase over the 20th century indicating less nutrient limiting conditions.

**p. 9-10, line 276-277: replace with "The mixed layer nutrient limitation patterns as estimated from NUTLIM and N/P ratios are shown in Figure 9."**

We have revised the manuscript in accordance with your comment.

**Figure 9.: In the figure it seems as if sometimes both N and P limitation overlap. Is this simply due to the resolution of the figure? It might be helpful to plot the actual N:P ratios in the lower panel.**

Simultaneous N and P limitation is not possible. The appearance is solely due to the size of the rings in the figure. We have followed your recommendation and added a plot showing the actual N/P in the lower right panel.

**p. 10, line 279: replace with "during the first part of the run is consistent with the studies on pre-industrial conditions" and give the references for the studies referred to.**

We have revised the manuscript in accordance with your comment.

**Figures 10, 11, 12, 14, 15, 16, 17: Please explain the meaning of arrow orientations in more detail (i.e. legend in Fig. 10 does not explain how to interpret the angle, the first mention of it is on p. 11, lines 314-315).**

We have added an explanation to the arrow orientations in the captions of Figs. 10, 11, 12, 14, 15, 16 and 17.

**p. 11, line 313,316: replace loads with concentrations.**

We have revised the manuscript in accordance with your comment.

**p. 11, line 313 and elsewhere: What does DIN refer to? Is ammonium included like in legend of Fig. 7? Ammonium should not be included in DIN.**

Ammonium is included in DIN. If there is a reason for a different definition, please provide us with a reference.

**p. 11, lines 320-321: sentence is confusing. Figure 15 does not show large differences between phosphate and DIN.**

The figure shows differences for periodicities between 1 and 16 yrs. We have clarified in the text the periodicity interval that we refer to.

**p. 11, line 322: why would low salinity indicate stronger mixing?**

Periods with large freshwater supply and low mixed layer salinity results in a weakening of the halocline as deep water salinity decreases faster than mixed layer salinity. The weakened halocline in turn leads to increased deep mixing.

We have added a reference that had been missed in the manuscript version displaying the changes made to the manuscript.

**p. 11, line 332: could be replaced with "The mixed layer temperature in the Baltic proper has increased..."**

We have revised the manuscript in accordance with your comment.

**p. 12, lines 366-367: can be replaced with "biomass of individual phytoplankton groups increased to such an extend that inter-annual variations are small compared to the seasonal signal**

**in primary productivity...” Primary production is not discussed in the manuscript. What analysis (results) supports this statement?**

This is an error. We have changed primary production to phytoplankton biomass.

**p.13, lines 380-381: I am not sure what the message here is: does the estimate of nutrient limitation using N:P ratios better reproduce field observations as compared to the NUTLIM scheme?**

Nutrient limitation as calculated from N/P ratios is not directly comparable to the NUTLIM concept. NUTLIM is basically an efficiency, mapping a 3d space made up of $PO_4^{3-}$, $NO_3^-$ and $NH_4^+$ concentrations onto a value between 0 and 1. Limitations from N/P ratios meanwhile, are a 2d mapping from $PO_4^{3-}$ and DIN to a boolean variable. The more prevalent phosphate limitation in the model is thus not a manifestation of incorrect N/P ratios. Rather, it reflects the difference between the NUTLIM concept and N/P ratios.

We have added this comment on lines 327-330 in the manuscript.

[revised manuscript text omitted]

---

## Author Response (AR4)

**Authors comment**

Again, a great thanks to the associate editor and the two referees for their work throughout the review process that greatly improved the manuscript.

Some confusion around the atmospheric nutrient forcing lead to a mistake in the previous version of the manuscript. The atmospheric nutrient input has no specific separation between organic and inorganic nutrients and all is considered bioavailable. A comment on this and a new reference have been added to the new manuscript version.

[revised manuscript text omitted]